# The importance of particle size distribution and internal structure for triple-frequency radar retrievals of the morphology of snow

Shannon L. Mason[1,2], Robin J. Hogan[3,1], Christopher D. Westbrook[1], Stefan Kneifel[4],
Dmitri Moisseev[5,6], and Leonie von Terzi[4]

[1]University of Reading, Reading, UK
[2]National Centre for Earth Observation, Reading, UK
[3]European Centre for Medium-range Weather Forecasts, Reading, UK
[4]University of Cologne, Cologne, Germany
[5]University of Helsinki, Helsinki, Finland
[6]Finnish Meteorological Institute, Helsinki, Finland

**Correspondence:** Shannon L. Mason (s.l.mason@reading.ac.uk)

**Abstract.** The accurate representation of ice particles is essential for both remotely-sensed estimates of clouds and precipitation and numerical models of the atmosphere. As it is typical in radar retrievals to assume that all snow is composed of aggregate snowflakes, both denser rimed snow and the mixed-phase cloud in which riming occurs may be under-diagnosed in retrievals, and therefore difficult to evaluate in weather and climate models. Recent experimental and numerical studies have yielded

methods for using triple-frequency radar measurements to interrogate the internal structure of aggregate snowflakes, and to distinguish more dense and homogeneous rimed particles from aggregates.

     In this study we investigate which parameters of the morphology and size distribution of ice particles most affect the triple-frequency radar signature and must therefore be accounted for in order to carry out triple-frequency radar retrievals of snow. A range of ice particle morphologies are represented, using a fractal representation for the internal structure of aggregate

snowflakes, and homogeneous spheroids to represent graupel-like particles; the mass- and area-size relations are modulated by a density factor. We find that the PSD shape parameter and the parameters controlling the internal structure of aggregate snowflakes both have significant influences of triple-frequency radar signature, and are at least as important as that of the density factor. We explore how these parameters may be allowed to vary in order to prevent triple-frequency radar retrievals of snow from being over-constrained, using two case studies from the Biogenic Aerosols—Effects of Clouds and Climate (BAECC)

2014 field campaign at Hyytiälä, Finland. In a case including heavily rimed snow followed by large aggregate snowflakes, we show that triple-frequency radar measurements provide a strong constraint on the PSD shape parameter, which can be estimated from an ensemble of retrievals; however, resolving variations in the PSD shape parameter has a limited impact on estimates of snowfall rate from radar. Particle density is more effectively constrained by the Doppler velocity than triple-frequency radar measurements, due to the strong dependence of particle fallspeed on density. Due to the characteristic signatures of

aggregate snowflakes, a third radar frequency is essential to effectively constraining the size of large aggregates. In a case featuring rime splintering, differences in the internal structures of aggregate snowflakes are revealed in the triple-frequency radar measurements. We compare retrievals assuming different aggregate snowflake models against in situ measurements at

the surface, and show significant uncertainties in radar retrievals of snow rate due to changes in the internal structure of aggregates. The importance of the PSD shape parameter and snowflake internal structure to triple-frequency radar retrievals of snow highlights that the processes by which ice particles interact may need to be better understood and parameterised before triple-frequency radar measurements can be used to constrain retrievals of ice particle morphology.

## 1 Introduction

Remotely-sensed estimates of ice clouds and snow from spaceborne radars inform our understanding of key components of the global water and energy cycles. Both retrieval algorithms and numerical weather and climate models rely on a representation of the ice particle size distribution (PSD) and morphology, which are functions of the microphysical processes by which ice

particles form, interact and grow. The processes of deposition, aggregation and riming may all contribute to the formation of a snowflake, and this growth history is encoded in the morphology of an ice particle. As riming requires the interaction of precipitating ice with supercooled liquid droplets in mixed-phase cloud layers which are difficult to diagnose, it has long been assumed that the majority of snow falls as unrimed aggregate snowflakes (Langleben, 1954). However, recent global active remote-sensing suggests that mixed-phase clouds are frequently associated with snow, especially over the ocean (Battaglia and

Delanoë, 2013). Nevertheless, it has been typical to assume a fixed representation of the morphology and size distribution of ice particles, usually derived from measurements of unrimed aggregates (e.g. Delanoë and Hogan, 2008; Hogan et al., 2012). This means that many radar retrievals of snow do not represent particles which have grown by riming, a process which can contribute a significant fraction of the mass of snow (Mosimann, 1995; Grazioli et al., 2015; Tiira et al., 2016; Moisseev et al., 2017; von Lerber et al., 2017).

Radar retrievals that allow variation in the morphology of ice particles are therefore of significant interest, and the development of novel retrieval methods has been facilitated by measurement campaigns combining multiple-frequency radars and other remote-sensing instruments with in situ measurements of snow particle properties (e.g. Szyrmer and Zawadzki, 2014a; Petäjä et al., 2016). Doppler velocity measurements of snow have been used to constrain variations in particle density by which riming, and the mixed-phase cloud in which it occurs, can be diagnosed (Mosimann, 1995; Szyrmer and Zawadzki,

2014a, b; Mason et al., 2018). Variations in the density of ice particles are also related to changes in their shape and structure— whether due to aggregation, riming or a combination of processes—which are reflected in their radar backscatter cross-sections (Leinonen and Szyrmer, 2015). Mason et al. (2018) formulated a single parameter modulating the density, shape and structure of ice particles, which was retrieved in single- and dual-frequency Doppler radar retrievals, and was chiefly constrained by the mean Doppler velocity. This representation was based on a large database of ice particle mass- and area-size relations and

insights from remotely-sensed and in-situ measurements of snow events (Kneifel et al., 2015), but how the parameters control-

ling the size distribution and morphology of snow particles relate to one another and their radar scattering characteristics is not yet fully understood.

The relation between ice particle morphology and triple-frequency radar measurements emerges in a comparison of models for the radar backscatter cross-sections of ice particles (Kneifel et al., 2011). The triple-frequency radar 'signature' consists of radar measurements at three frequencies spanning the millimeter wavelength range, the two dual-wavelength ratios (DWRs) of which reveal information about non-Rayleigh scattering from larger snowflakes. Typically radars at 95, 35 and a third frequency between 3 and 15 GHz are used (e.g. Kneifel et al., 2015; Leinonen and Szyrmer, 2015; Barrett et al., 2019). The triple-frequency radar signature has provided a succinct means of evaluating the applicability of spheroidal particles as models for fractal aggregate snowflakes (Leinonen et al., 2012), revealing the fractal dimension of observed aggregates (Stein et al., 2015), and exploring their microphysical structure (Leinonen and Moisseev, 2015). Triple-frequency radar observations of rimed snow combining ground-based radar and in situ measurements (Kneifel et al., 2015) showed the apparent influence of increasing particle density on the triple-frequency radar signature, while a modelling study of the combined effects of growth by aggregation and riming related triple-frequency measurements to microphysical processes (Leinonen and Szyrmer, 2015). Combining triple-frequency and Doppler velocity information, Kneifel et al. (2016) used triple-frequency Doppler spectra to identify the spectral signatures of rimed and unrimed snow. The insights provided by triple-frequency radar techniques have contributed to the development of expanded scattering databases for representing and evaluating a wide range of ice particles (Kneifel et al., 2018).

The strong numerical and observational evidence that the triple-frequency radar signature reflects the density and structure of snow particles suggests the potential for retrievals in which some morphological parameters are constrained by triple-frequency radar measurements. Triple-frequency radar retrievals have been demonstrated in which the structure and density of ice particles are allowed to vary, but where the PSD Shape is assumed constant (Leinonen et al., 2018b; Tridon et al., 2019); however, it is not yet clear which properties of ice particle are best constrained by the triple-frequency radar measurements. Leinonen et al. (2018b) found that a triple-frequency radar retrieval of the mass-size relation of ice particles did not differ significantly from that of a dual-frequency retrieval, suggesting that the problem is over-constrained. It has been shown that the higher-order moments of the Doppler spectra from multiple-frequency radars can be used to reduce the uncertainty of retrievals of ice clouds (Maahn and Löhnert, 2017); however, triple-frequency radar signatures are most distinct for larger precipitating particles. To our knowledge a retrieval assimilating both triple-frequency radar reflectivity factors and mean Doppler velocity to estimate the properties of snow has not yet been described. This approach should have the advantages of constraining particle density with Doppler velocity (as in Mason et al., 2018), while using triple-frequency radar signatures to constrain some additional parameter affecting the microphysical properties or size distribution of precipitating ice particles.

In this study we explore the potential for a triple-frequency Doppler radar retrieval using the optimal estimation framework Cloud Aerosol and Precipitation from mulTiple Instruments using a VAriational TEchnique (CAPTIVATE; Mason et al., 2018). In Section 2 we briefly describe the key components of the radar forward-model, and the remotely-sensed and in situ data used to perform and evaluate the retrieval. In Section 3 we explore how parameters controlling the PSD and particle properties affect the forward-modelled triple-frequency radar signatures of aggregate snowflakes and graupel-like particles. We then apply these

insights to triple-frequency radar measurements from the Biogenic Aerosols—Effects of Clouds and Climate (BAECC) field campaign in Hyytiälä, Finland in 2014 (Petäjä et al., 2016). In Section 4 we use our particle models to represent the triple-frequency radar signatures of rimed and unrimed snow regimes based on their structure, density and PSD shape parameter. We demonstrate a triple-frequency Doppler radar retrieval of the PSD shape parameter and particle morphology, and evaluate remotely-sensed estimates against in situ measurements at the surface. In Section 5 we consider the triple-frequency radar signatures from a case study featuring secondary ice production due to rime splintering, with a focus on the presence of large aggregates with distinct triple-frequency radar signatures. Finally, we summarise our conclusions in Section 6.

## 2 Methods and data

We first describe the state and observation variables in the retrieval framework (Section 2.1), then outline the radar measurements assimilated into the retrieval and in situ measurements against which the retrieval is evaluated (Section 2.2).

### 2.1 Radar forward model and retrieval algorithm

We use the optimal estimation retrieval algorithm Cloud Aerosol and Precipitation from Multiple Instruments using a Variational Technique (CAPTIVATE) described by Mason et al. (2018). This framework has been developed for retrievals with the Doppler cloud profiling radar aboard EarthCARE (Illingworth et al., 2015), but is configurable for multiple radar instruments in ground-based and airborne, as well as spaceborne, applications. Here we describe the major components of the CAPTIVATE radar forward-model pertinent to ice and snow.

The ice PSD is given by

$$N(D) = N_w F(D/D_0, \mu), \tag{1}$$

where $N$ is the number of particles of maximum dimension $D$, $N_w$ is the normalized number concentration, and $F$ is the normalized gamma distribution (Testud et al., 2001; Illingworth and Blackman, 2002; Delanoë et al., 2005), which is a function of the median volume diameter ($D_0$) and the shape parameter of the PSD ($\mu$). Modified 'universal' PSDs formulated to address the need for non-exponential distributions in ice clouds (Delanoë et al., 2005; Field et al., 2005, 2007) are also implemented in CAPTIVATE (Mason et al., 2018), but in this study Gamma PSDs are used in order to explore the effects of accounting for PSDs broader ($\mu < 0$) and narrower ($\mu > 0$) than the exponential. While the majority of triple-frequency radar studies assume an exponential PSD (i.e. $\mu = 0$), in situ measurements of snow at Hyytiälä suggest Gamma PSD shape parameters vary in the range $-2 < \mu < 5$ (Fig. 15 in Tiira et al., 2016).

The morphology of ice particles is represented by parameters controlling their microphysical structure, density, and shape. Approximations to the microphysical structure of ice particles are used to calculate the radar backscatter cross-section ($\sigma$) of each particle. We use two methods to represent the range of particle structures from aggregates snowflakes to graupel. The scattering cross-section of low density particles such as aggregates can be represented using the Rayleigh-Gans approximation, wherein the interactions between dipoles within the particle are neglected, and the electric field experience by each dipole is

equal to the incident field. In the Self-Similar Rayleigh Gans Approximation (SSRGA; Hogan and Westbrook, 2014; Hogan et al., 2017) the radar backscatter cross-section of a low-density particle deviates from Rayleigh scattering according to

$$\phi_{\text{SSRGA}}(kD) = \frac{\pi^2}{4} \left( \cos^2(kD)A(kD,\kappa) + \sin^2(kD) \sum_{j=1}^{n} B(j,\zeta,\beta,\gamma) \right), \tag{2}$$

where $k = 2\pi/\lambda$ is the wavenumber at radar wavelength $\lambda$ and $D$ the maximum dimension of the particle. The effect of the average particle geometry $A$ is a function of the kurtosis $\kappa$ of the distribution of mass about the centre of the particle. The effect of the internal structure is given by the integral over a power spectrum representing random fluctuations about the average particle geometry,

$$B = \beta\zeta_j(2j)^{-\gamma} \left[ \frac{1}{(2kD + 2\pi j)^2} + \frac{1}{(2kD - 2\pi j)^2} \right] \tag{3}$$

where $j$ is the index of the wavenumber. The power law prefactor $\beta$ represents the amplitude of fluctuations about the average particle geometry, while the power law exponent $\gamma$ controls the relative importance of fluctuations at small scales. The scaling factor $\zeta_j$ was introduced in Hogan et al. (2017) to reduce the amplitude of the internal structure at the largest scale (i.e. $\zeta_1 \leq 1$, then $\zeta_j = 1$ for $j > 1$).

A fit to aggregates of bullet rosettes from the aggregation model of Westbrook et al. (2004) was derived by Hogan et al. (2017). Based on the observation that aggregates of most monomers (e.g. dendrites, plates and columns) had similar statistics, this model has been applied to CAPTIVATE retrievals of aggregate snowfall (e.g. Mason et al., 2018); however, in Section 3 we will also consider the effect of variations in snowflake structure on the triple-frequency radar signature. As the Rayleigh-Gans approximation applies to low-density particles in which the interactions between dipoles can be neglected, denser heavily rimed aggregates and graupel particles are represented using the T-matrix approximation to 'soft spheroids' composed of homogeneous mixtures of ice and air (e.g. Hogan et al., 2012). This provides a good approximation to graupel, but is known to under-estimate the backscatter from aggregate snowflakes (Leinonen and Szyrmer, 2015).

The density factor $r$ varies both the prefactor and exponent of the ice particle mass-size relation

$$m(D) = aD^b \tag{4}$$

between that of the "aggregates of unrimed bullets, columns and side-planes" of Brown and Francis (1995) at $r = 0$, and that of spheroids of solid ice at $r = 1$. Mason et al. (2018) showed that this parameter allowed for simplified representation of a broad range of measured mass-size relations for particles along a continuum from unrimed aggregates to rimed snow, graupel, and hail. Finally, the volumetric shape of all particles is defined by the axial ratio ($AR$) of horizontally-aligned oblate spheroids.

We note that in nature the structure, density and shape of snow particles are not independent. As discussed in Mason et al. (2018), the fractal or homogeneous distribution of mass through the volume of a particle is closely related to its density: the fractal structure of aggregates snowflakes is characterised by a mass-size relation with an exponent close to $b = 2$, while more homogeneous graupel and hail particles have mass-size relations with $b$ approaching 3. The riming process by which aggregates accrete mass to become more graupel-like is also known to change the particle shape, with axial ratios increasing from more oblate snowflakes ($AR \approx 0.6$) to rounder graupel particles ($AR > 0.8$) (Li et al., 2018).

### 2.1.1 State vector

As described in Mason et al. (2018), CAPTIVATE has been developed for radar-lidar-radiometer synergy from the upcoming EarthCARE satellite, and the retrieval of ice and snow follows the work of Delanoë and Hogan (2008) for radar-lidar synergy. The state vector for a vertical profile is:

$$\mathbf{x} = \begin{pmatrix} \ln \alpha_{\mathrm{v}} \\ \ln N_0' \\ r' \end{pmatrix}, \tag{5}$$

where the state variables, the visible extinction coefficient $\alpha_{\mathrm{v}}$ in the geometric optics approximation and the primed number concentration $N_0'$, are chosen so that prior estimates can be made as a function of atmospheric temperature (see Fig. 3 of Delanoë and Hogan, 2008). Retrieving both of these terms provides sufficient degrees of freedom to derive two parameters of the PSD, the median volume diameter and normalized number concentration; these more physically meaningful values, rather than the state variables, are reported in this study. The natural logarithms of most parameters are used to avoid non-physical negative values and to improve convergence. The final state variable is the density index $r'$, a function of the density factor such that $r'$ is defined at all real values (Mason et al., 2018). To reduce the effect of measurement noise on the retrieval, the retrieved state variables through the vertical profile are represented as the basis functions of a cubic spline (Hogan, 2007). The PSD shape parameter, axial ratio and chosen radar backscatter approximation are configurable at runtime, but are assumed constant within each retrieval.

### 2.1.2 Measurement vector

The radar reflectivity factor at frequency $f$ is given (in linear units) by

$$Z_f = \frac{\lambda_f^4}{\pi^5 |K_w|^2} \int\limits_0^\infty \sigma_f(D) \, N(D) \, dD \tag{6}$$

where $\lambda_f$ is the radar wavelength, $K_w$ is the dielectric factor of water at cm wavelengths, and $\sigma_f(D)$ is the backscatter cross-section for a particle of maximum dimension $D$ at the radar frequency. The dual-wavelength ratio (DWR) between frequencies $f_1$ and $f_2$ is then $\mathrm{DWR}_{f_1 - f_2} = Z_{f_1} / Z_{f_2}$. Both radar reflectivity and DWR are reported in dB unless otherwise stated. While DWR quantities are reported here, the radar reflectivity factors at each frequency are used in the measurement vector.

Mean Doppler velocity is given by

$$V_f = \frac{\int_0^\infty v(D) \, \sigma_f(D) \, N(D) \, dD}{\int_0^\infty \sigma_f(D) \, N(D) \, dD}, \tag{7}$$

where the terminal velocity of an ice particle $v(D)$ assumes negligible vertical air motion, and positive values of mean Doppler velocity are toward the surface.

The measurement vector for $n$ radar frequencies is therefore given by

$$\mathbf{y} = \begin{pmatrix} Z_{\mathrm{f}_1} \\ \vdots \\ Z_{\mathrm{f}_n} \\ V_{\mathrm{f}_1} \\ \vdots \\ V_{\mathrm{f}_n} \end{pmatrix}. \tag{8}$$

In this paper we perform retrievals using two or three radar frequencies, and use mean Doppler velocity at one radar frequency.

## 2.2 Radar and in situ measurements

Atmospheric Radiation Measurement second mobile facility (AMF2) Doppler radars at 10, 35 and 95 GHz were deployed at Hyytiälä, Finland during the BAECC 2014 field campaign (Petäjä et al., 2016). Radar measurements used here are at ~2 s temporal and ~30 m vertical resolution, and radar reflectivity factors have been corrected for gaseous and liquid attenuation. The specifications of the AMF2 radars, and their colocation, calibration and attenuation correction for triple-frequency radar measurements during BAECC 2014 are described in Kneifel et al. (2015). Due to a mispointing of the 95 GHz radar during BAECC, the mean Doppler velocity for that radar is not used in this study.

In situ measurements of snow at the surface are provided by the Particle Imaging Package (PIP) video disdrometer (Newman et al., 2009). While the temporal resolution of PIP measurements is 1 minute, estimates of parameters of the PSD and particle properties are made over 5 minute intervals in order to increase the statistical sampling during BAEC while still resolving changes in the properties of snowfall at the surface, as described in von Lerber et al. (2017) (also Moisseev et al., 2017; Tiira et al., 2016). The method of moments is used to estimate the parameters of the Gamma distribution from the measured PSD (Moisseev and Chandrasekar, 2007).

Due in part to the scanning strategy of the 35 and 10 GHz radars during BAECC, zenith-pointing measurements from all three radars coincident with sufficient snowfall measured by PIP at the surface were rare during the observation period. The case studies consists of approximately 25 minutes of zenith-pointing triple-frequency Doppler radar data between 22:53 and 23:18 UTC on 21 February 2014 (Section 4), and one hour between 00:00 and 01:00 UTC on 16 February 2014 (Section 5). Radar measurements are complemented by contemporary radiosonde profiles and numerical weather prediction analyses which provide thermodynamic information for the retrieval.

## 3 Parameters affecting the triple-frequency radar signature

In this section we explore the effects of ice particle morphology and size distribution on the triple-frequency radar signature. As described in Section 2.1, the representation of ice particles can be controlled by parameters for their microphysical struc-ture, density and shape. Two approximations to the radar backscatter cross-section are used to represent the range of particle

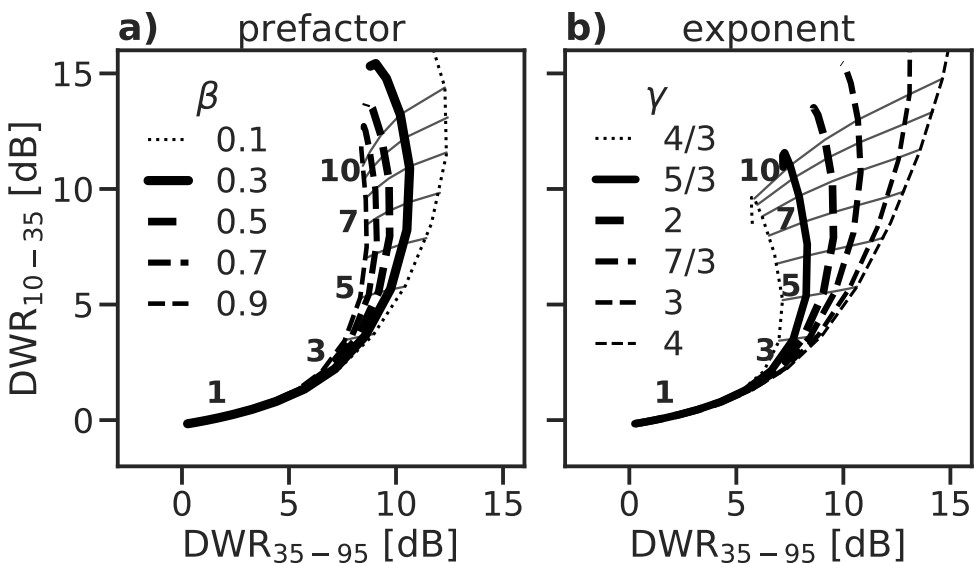

**Figure 1.** Forward-modelled triple frequency radar signatures for fractal particles represented by the SSRGA, with a range of values of (a) the power law prefactor $\beta$, where the exponent is constant ($\gamma = 2.0$); and (b) the power law exponent $\gamma$, where the prefactor is constant ($\beta = 0.55$). The other SSRGA parameters used are for the aggregates of bullet rosettes of Hogan et al. (2017), given in Table 1. Lateral lines denote increments of median volume diameter, labelled in millimetres. All particles are assumed to have an exponential PSD ($\mu = 0$), a density factor of $r = 0$, and a particle axial ratio of $AR = 0.6$.

structures: low-density aggregate snowflakes are represented as fractal particles, the complex internal of structures of which are controlled by the coefficients of a power law; while dense graupel-like particles are modelled as homogeneous spheroids. For both fractal and homogeneous particles, the density factor and the axial ratio control the mass-size relation and the shape of the volume enclosing the particle. The final parameter is the PSD shape parameter, which is independent of particle morphology but affects the relative weighting given to particles across the size spectrum. Most studies of triple-frequency radar signatures of ice particles have assumed exponential PSDs (Kneifel et al., 2011; Leinonen et al., 2011; Kneifel et al., 2015; Leinonen and Moisseev, 2015; Leinonen and Szyrmer, 2015).

We first consider the effect of variations in the internal structure of aggregates as represented in the SSRGA. In Figure 1 we show the sensitivity of the triple-frequency signature to the coefficients of the power spectrum representing the fluctuations about the average particle. Dual-wavelength ratios are enhanced by strong peaks in backscatter ratios at particle sizes where the radar backscatter is affected by constructive or destructive interference at one of the wavelengths (illustrated later in Fig. 4a). These spectral features are strongest in the radar backscatter cross-sections of soft spheroids (Fig. 9c of Hogan et al., 2017). The SSRGA superimposes random smaller-scale structure onto the average particle structure, introducing random interference

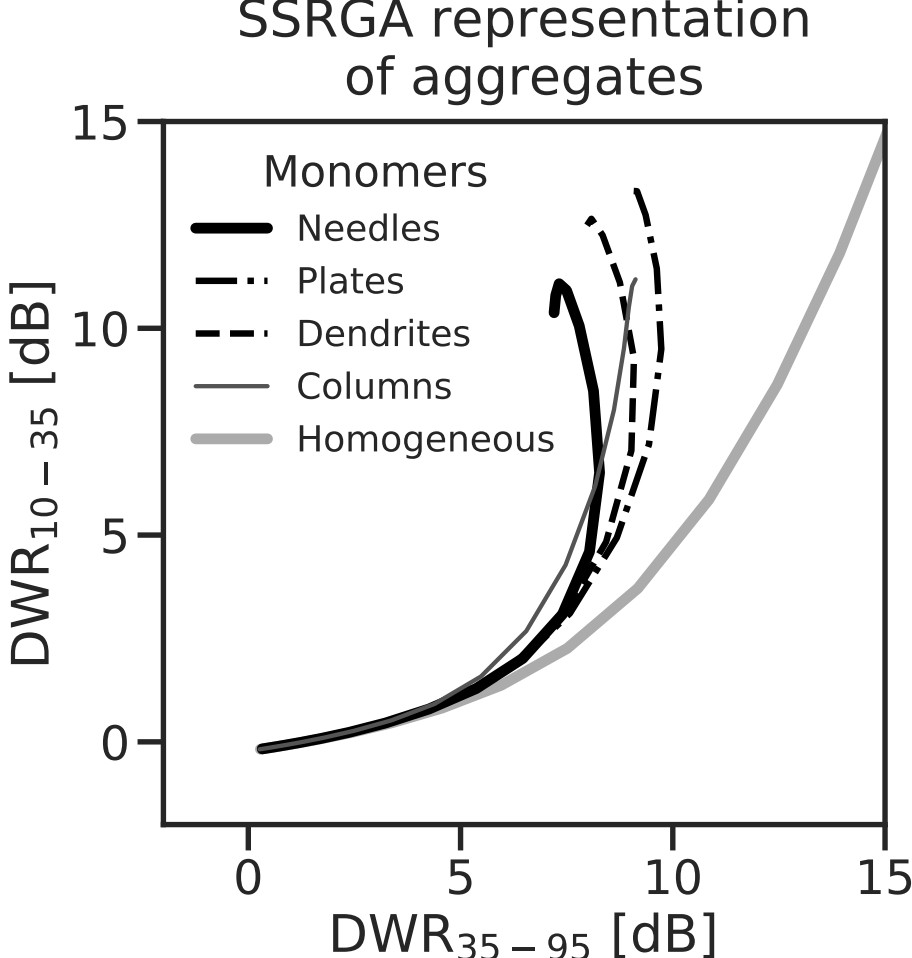

**Figure 2.** Triple-frequency radar signatures for SSRGA representations of four aggregates of different monomers (Table 1), and a representation of homogeneous particles ($\beta = 4$).

which smooths the resonant features over the spectrum, and reduces the maximum dual-wavelength ratios. The dual-wavelength ratios are reduced by either increasing the amplitude of the fluctuations (increasing the prefactor $\beta$), or by increasing the significance of structure at small scales (decreasing the exponent $\gamma$). Reducing the power law exponent to represent aggregates with a greater degree of structure at small scales also results in a more pronounced hook feature. Conversely, a power law exponent of $\gamma = 4$ is associated with the homogeneous distribution of mass at or below the scale of monomers in aggregates (Hogan and Westbrook, 2014); the triple-frequency radar signatures at this limit, or where the amplitude of the fluctuations $\beta$ approaches zero, come to resemble those of homogeneous spheroids with no hook feature (Fig. 2).

To identify the variability in these parameters for a range of aggregate snowflakes, we fit SSRGA parameters to simulated aggregates comprising a range of monomer types and size distributions. The various aggregates were generated using the

| Monomer | $\kappa$ | $\beta$ | $\gamma$ | $\zeta_1$ |
|---|---|---|---|---|
| Bullet rosettes (Hogan et al., 2017) | 0.09 | 0.55 | 2.0 | 0.28 |
| Plates | 0.18 | 0.8 | 2.1 | 0.10 |
| Dendrites | 0.20 | 0.6 | 1.8 | 0.13 |
| Columns | 0.22 | 1.96 | 2.15 | 0.09 |
| Needles | 0.25 | 0.76 | 1.66 | 0.10 |

**Table 1.** SSRGA parameters fit to simulated aggregates of a range of monomers.

aggregation code presented in Leinonen and Moisseev (2015), which assumes an inverse exponential distribution of monomer sizes from which the monomers are sampled. In total more than 30,000 aggregates of dendrite, plate, needle, and column monomers were generated covering maximum sizes up to 2 cm. For this, the characteristic size of the monomer distribution was varied from 0.2 to 1.0 mm, with minimum monomer size of 0.1 mm and maximum size of 3 mm. Each aggregate consisted of up to 1000 monomers. SSRGA fits to aggregates of the range of monomers (Table 1) have similar characteristic triple-frequency signatures (Fig. 2). The notable exception is aggregates of needles, for which the triple-frequency signature tends toward lower values of $DWR_{35-95}$, and the SSRGA power law exponent is lower, around $\gamma = 5/3$ rather than $\gamma \approx 2$. The power law prefactor for aggregates of needles denotes a relatively high degree of structure, with $\beta \approx 0.76$. These results are consistent with Leinonen and Moisseev (2015, their Fig. 3e), who used the discrete dipole approximation to generate the triple-frequency radar signatures of a range of modelled aggregate particles and found that aggregates of large needles had characteristically lower values of $DWR_{35-95}$ compared with aggregates of other monomers. To encompass this range of triple-frequency radar signatures for aggregates, we employ SSRGA fits to two genres of fractal particles: aggregates of bullet rosettes to represent most snowflakes (Hogan et al., 2017); and aggregates of needles.

Next we compare the triple-frequency radar signatures for aggregates of needles (Fig. 3 a–c), aggregates of bullet rosettes (Fig.3d–f), and homogeneous spheroids (Fig. 3 g–i) for a range of values of the PSD shape parameter, density factor and axial ratio. Qualitatively, we note that the dual-wavelength ratios achieved by varying these parameters cover a similar range to the statistics of triple-frequency radar measurements (Dias Neto et al., 2019). The triple-frequency radar signatures for the fractal particles exhibit maxima in $DWR_{35-95}$ between 8 and 12 dB for aggregates of bullet rosettes, and lower values around 7 to 8 dB for the aggregates of needles. After reaching their maxima at median volume diameters around 5 to 7 mm, the triple-frequency radar signatures double back such that most curves are non-unique in $DWR_{35-95}$ while $DWR_{10-35}$ continues to increase with median particle size. This hook feature in the triple-frequency radar signature is characteristic of aggregates in theoretical and observational studies (e.g. Kneifel et al., 2015), and the concavity of the hook feature increases with both PSD shape parameter for fractal particles (Fig. 3 a & d) and density factor (Fig. 3 b & e). This saturation of $DWR_{35-95}$ for aggregates is expected to pose a challenge for dual-frequency retrievals with 35 and 95 GHz radars, such that it will be difficult to differentiate moderately-sized from very large aggregates. This effect would not be aided by the assimilation of mean Doppler velocity, because the terminal fall speed of the largest unrimed aggregates also saturates (e.g. Mitchell and Heymsfield, 2005). The

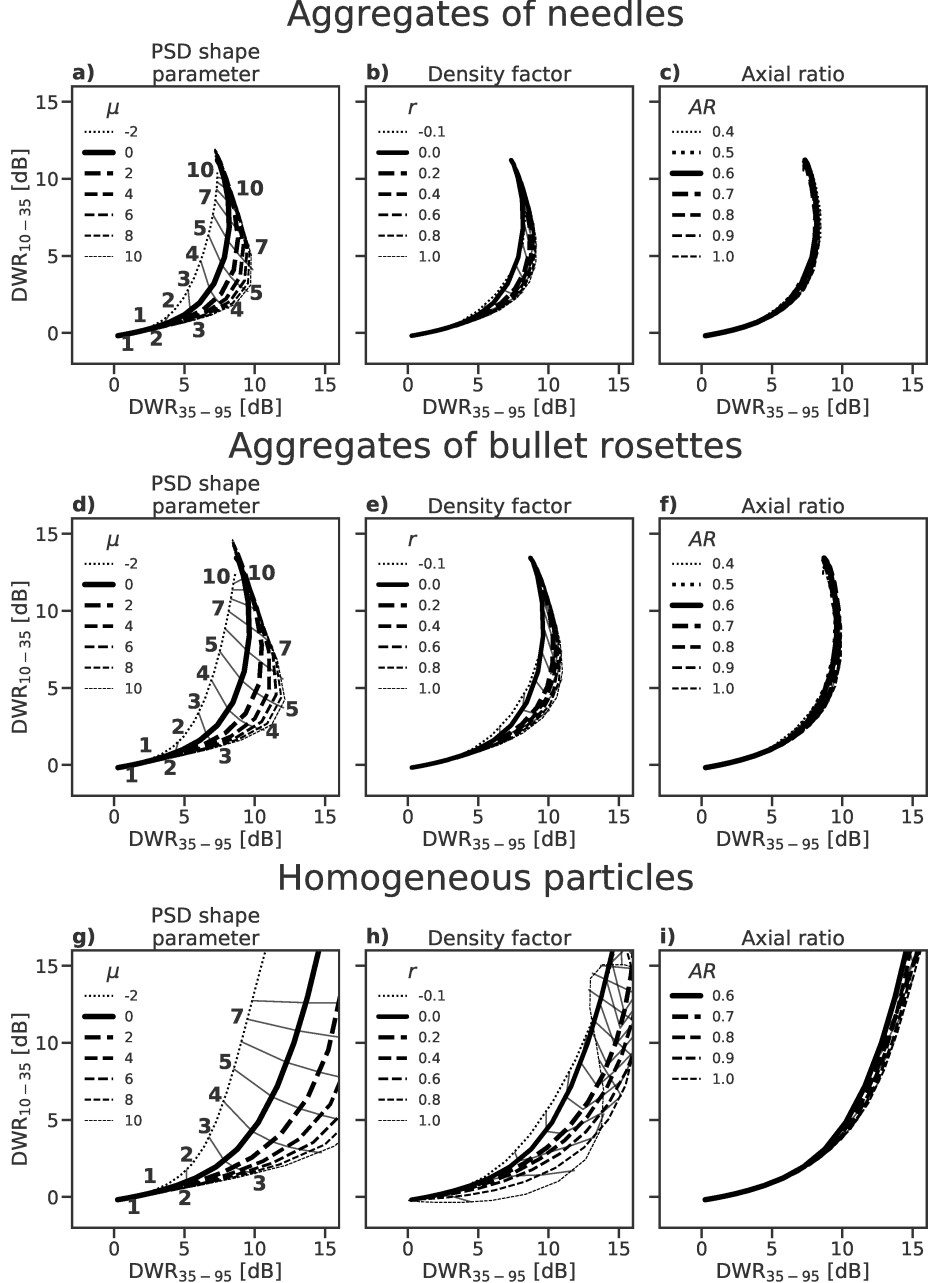

**Figure 3.** Simulated triple-frequency radar signatures for distributions of (a–c) fractal particles represented using SSRGA and (d–f) homogeneous ("soft") spheroids. Black lines illustrate radar signatures for a spectrum of PSDs with median volume diameters (values indicated in panels a and d) for different values of (a, d & g) PSD shape parameter, (b, e & h) density factor, and (c, f & i) axial ratio. Lateral lines denote increments of median volume diameter in millimeters.

triple-frequency radar signature is relatively insensitive to the axial ratio of the spheroidal particles (Fig. 3 c & f). The triple-frequency signatures differ with particle density and PSD shape parameter between median volume diameters of 2 mm and 8 mm, but converge outside this range. Increasing the PSD shape parameter results in greater maximum values of $DWR_{35-95}$: while an exponential PSD reaches a maximum of around 9 dB at a median volume diameter around 7 mm, narrower PSDs reach $DWR_{35-95}$ of up to 12 dB at median volume diameters around 5 mm. A smaller range of triple-frequency signatures is attributable to changes in the density factor: low-density aggregates of bullet rosettes exhibit a maximum $DWR_{35-95}$ of around 9 dB at 7 mm median volume diameter, and increase to around 11 dB for denser particles with $r > 0.4$; the maximum $DWR_{35-95}$ of aggregates of needles increase from around 7 to 8 dB. We note that the SSRGA snowflake model is suited to low-density aggregate particles; at higher density factors the Rayleigh-Gans approximation is no longer valid due to the interaction between dipoles within the particle. An evaluation of the suitability of SSRGA to dense rimed particles is assessed in Leinonen et al. (2018a).

The flat signatures of homogeneous spheroids reach greater values of $DWR_{35-95}$ than the fractal particles, even at relatively small median volume diameters of 3 to 4 mm. The triple-frequency radar signature for homogeneous spheroids resembles that shown in Fig. 1 for aggregate particles with either a power law amplitude of $\beta = 0$ or a power law exponent of $\gamma = 4$. This similarity is maintained up to density factors around $r = 0.5$, where the interaction between dipoles in the homogeneous particles becomes non-negligible. Both PSD shape parameter and density factor affect a shift of the signature to the lower-right of the diagram and, unlike fractal particles, the signatures of homogeneous spheroids do not converge at large median volume diameters. Increasing the density factor of homogeneous spheroids increases $DWR_{35-95}$; however, the hook feature only becomes evident for high density factors at large median volume diameters. Notably, at very high density factors $DWR_{10-35} \approx$ 0 dB are maintained for values of $DWR_{35-95}$ up to around 10 dB: this characteristic signature of very dense graupel is not observed for very high PSD shape parameters.

Regardless of particle structure, the axial ratio (AR) of the particles has a relatively minor influence on the triple-frequency signature. We therefore maintain the assumption of $AR = 0.6$ for all particles in this study, but note that uncertainty in this value contributes to uncertainties in the retrieval in other respects. For example, Mason et al. (2018) found that assuming $AR = 0.8$ leads to a roughly 20% increase in retrieved ice water content when compared to $AR = 0.6$. In situ measurements show that heavily rimed and graupel particles tend to have higher axial ratios (e.g. Garrett et al., 2015; Li et al., 2018), so including this effect in future retrievals may help to constrain uncertainties in estimates of dense rimed snow.

While no one particle morphology can represent the wide range of measured triple-frequency radar signatures (Dias Neto et al., 2019), there is also significant ambiguity between the radar signatures of small particle of all types, especially for median volume diameters less than around 3 mm. This makes the application of triple-frequency radar retrievals especially relevant to snow, rather than cloud ice. In an attempt to encompass the continuum of particle types—acknowledging that the structure and density of ice particles are interrelated—Mason et al. (2018) formulated a hybrid representation which transitions from fractal particles at low density factors ($r < 0.2$) to represent unrimed and lightly rimed aggregates, to homogeneous spheroids at high densities ($r > 0.5$) representing graupel-like particles. Intermediate moderately rimed aggregates in the range $0.2 < r < 0.5$ ("hybrid particles" in Fig. 4 a) are represented by an external mixture of the backscatter cross-sections of the fractal and

homogeneous models. More detailed parameterisations of this continuum may be achieved using the representation of a range of aggregates with different degrees of riming (Leinonen and Szyrmer, 2015; Leinonen et al., 2018a), or fractal structures (Leinonen and Moisseev, 2015; Hogan et al., 2017), and should be the subject of further work.

We have shown that, based on simulated radar backscatter cross-sections, the PSD shape parameter has a greater influence on the simulated triple-frequency radar signature than the well-known effect of particle density. This result holds for the aggregate snowflakes modelled as fractal particles and for graupel-like particles modelled as homogeneous spheroids. The effect of the PSD shape parameter is independent of the particle morphology, but results from changing the relative weighting of different parts of the particle size spectrum. This can be shown using the dual wavelength ratio on a per-particle basis, which we call the dual backscatter ratio following Kneifel et al. (2016),

$$\mathrm{DBR}_{1,2} = \left(\frac{f_2}{f_1}\right)^4 \frac{\sigma_{f_1}(D)}{\sigma_{f_2}(D)}, \tag{9}$$

in which the spectral features correspond to the onset of non-Rayleigh scattering at the higher frequency (Kneifel et al., 2016). $\mathrm{DBR}_{10,35}$ and $\mathrm{DBR}_{35,95}$ for fractal particles (aggregates of bullet rosettes) and homogeneous spheroids with $r = 0$ and $AR = 0.6$ are compared in Fig. 4 a alongside the volume-weighted particle size distributions for selected PSDs (Fig. 4 b): the exponential PSD (Marshall and Palmer, 1948), and PSDs measured during the rimed and unrimed snow regimes of the 21 February 2014 case at Hyytiälä, Finland, which is considered in more detail in Section 4. The unrimed snow regime fits a broader Gamma PSD (e.g. $\mu = -1$), and the rimed snow a narrower PSD (e.g. $\mu = 5$). As radar measurements relate to PSD-weighted integrals of the radar backscatter spectra by (1), the triple-frequency signature is strongly influenced by the median volume diameter and shape parameters of the PSD, with the PSD shape parameter modulating the relative influence of spectral features close to the median volume diameter. As the median volume diameter approaches the onset of non-Rayleigh scattering at 95 GHz (around 3 mm), $\mathrm{DWR}_{35-95}$ increases, creating the initial horizontal part of the triple-frequency radar signatures of both fractal particles and homogeneous spheroids; correspondingly, at the onset of non-Rayleigh scattering at the 35 GHz (around 8 mm) there is a shift to larger values of $\mathrm{DWR}_{10-35}$, and a vertical uptick in the triple-frequency diagram. For fractal particles (dashed lines in Fig. 4 a) there is a clear distinction between the parts of the spectrum dominated by non-Rayleigh scattering at 95 and 35 GHz: the decrease in the 35–95 GHz backscatter cross-section ratio at median diameters greater than around 6–8 mm results in the "bending-back" part of the hook feature in the triple-frequency signature, which occurs at smaller median diameters for narrower PSD shape parameters (Fig. 3 a).

For homogeneous spheroids (solid lines in Fig. 4 a) the many narrow features of the backscatter cross-section ratio spectra are smoothed out when integrated across the PSD, resulting in a flatter triple-frequency radar signature. This illustrates how a narrower PSD will increase the weight given to particles closest to the median volume diameter, exaggerating the effects of nearby features of the radar backscatter spectra and deepening the hook feature characteristic of fractal particles as PSD shape parameter increases. Conversely, broader PSDs have the effect of smoothing over the spectral features, producing an earlier onset of high values of $\mathrm{DWR}_{10-35}$ and a shallower hook feature for fractal particles.

We may therefore add to the diagram for the triple-frequency radar signature proposed in Kneifel et al. (2015) to include the effects of the PSD shape parameter and the internal structure of aggregate snowflakes, in addition to the effects of particle

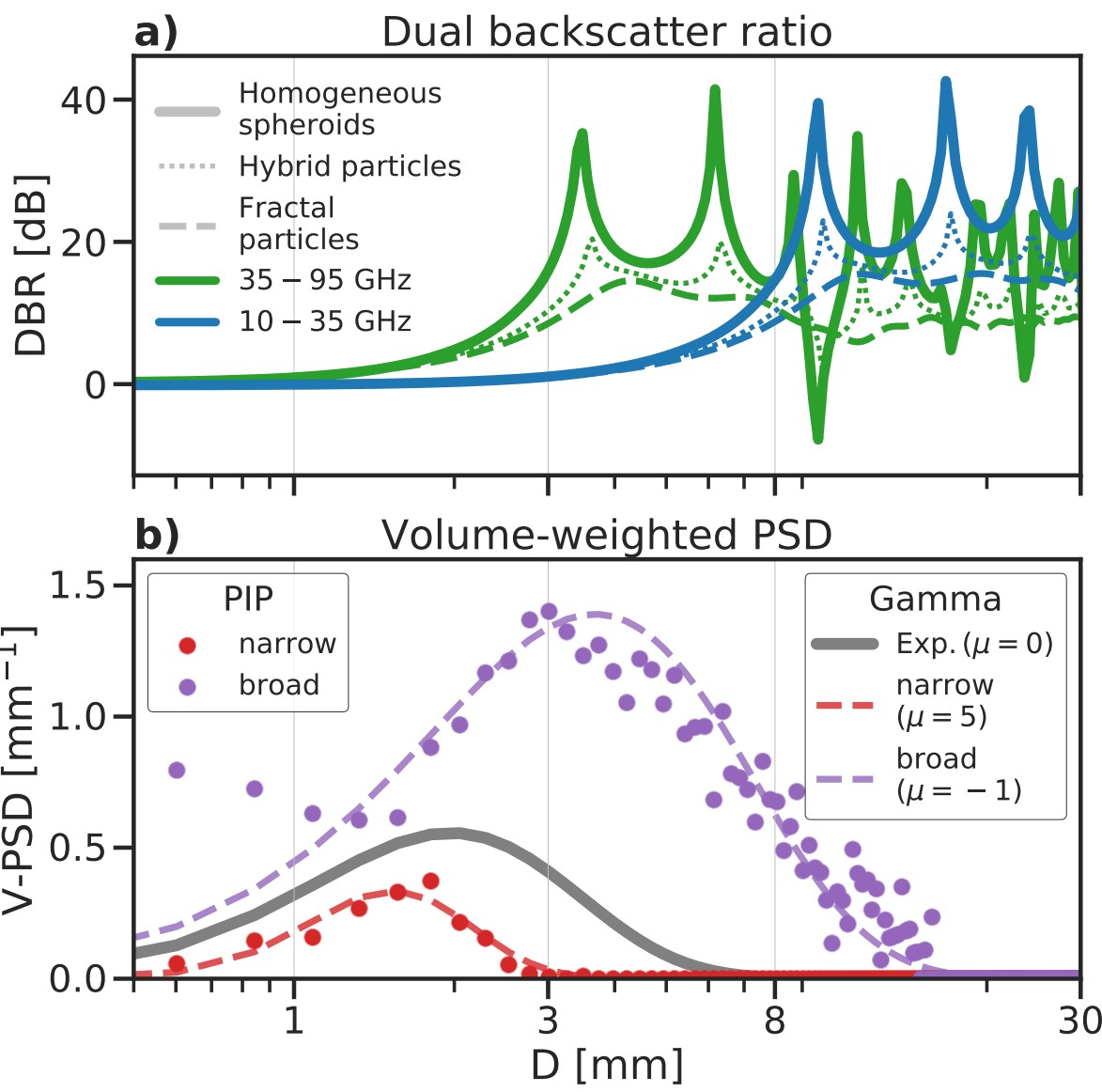

**Figure 4.** (a) Normalized dual backscatter ratios (DBR) at 10–35 GHz and 35–95 GHz bands for fractal particles (aggregates of bullet rosettes) and homogeneous spheroids, on the same size spectrum as (b) the volume-weighted PSD (V-PSD) measured in situ during the 21 February case study (Section 4) and Gamma PSDs fit to the same regimes, and an exponential PSD ($\mu = 0$). The backscatter cross-section ratios of the "hybrid" particles in (a) illustrate the transition between aggregates of bullet rosettes and homogeneous spheroid approximations for intermediate density factors between $0.2 < r < 0.5$, as described in Mason et al. (2018).

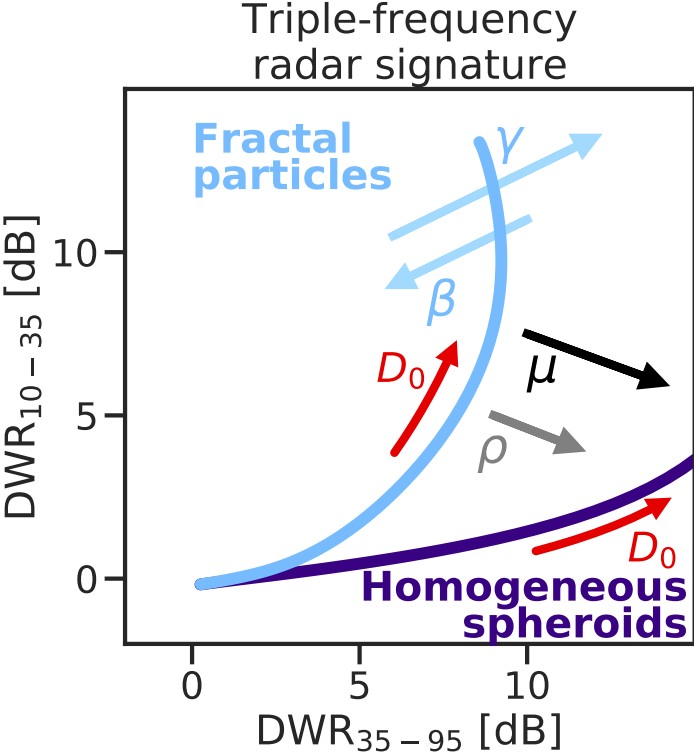

**Figure 5.** A schematic for the parameters affecting the triple-frequency radar signature of a spectrum of ice particles. In addition to the median volume diameter and density, the effects of the internal structure of aggregate particles and the PSD shape parameter are illustrated (cf. Kneifel et al., 2015).

density (Fig. 5). These reflect important insights into the interpretation of triple-frequency radar signatures: it is possible that the observed range of measurements in Kneifel et al. (2015) are attributable to the combined effects of increased particle density due to riming, of a narrower or broader PSD, and of variations in the internal structure of aggregate snowflakes. It is not apparent to what degree these parameters are independent of one another: Tiira et al. (2016) found only a weak relation between the bulk density and PSD shape parameter estimated from PIP measurements at Hyytiälä. Each of these effects represent an additional challenge when seeking to retrieve the morphology of particles from the triple-frequency radar measurements, a problem we will explore further in the next sections.

## 4  Retrievals of density factor and the PSD shape parameter: 21 February 2014 case study

Between 22:53 and 23:17 UTC on 21 February 2014 light snowfall dominated by compact graupel gave way to moderate snow characterised by larger unrimed aggregates (see Section 3.5.1 of Kneifel et al., 2015, for a detailed discussion of the meteorological context and in situ observations of this case). Mason et al. (2018) exploited Doppler velocity during a longer

time series of this event to constrain CAPTIVATE retrievals of the density factor, using single- and dual-frequency Doppler radar measurements. We divide the 25 minute case into rimed (22:53 to 23:03 UTC) and unrimed regimes (23:03 to 23:18 UTC) based on a combination of remotely-sensed (Fig. 6) and in situ data. Kneifel et al. (2015) showed that the onset of unrimed snow corresponded to the emergence of the distinct hook feature in the triple-frequency signature. This corresponds to the large values

of both $DWR_{35-94}$ and $DWR_{10-35}$ (Fig. 6b & c) near the surface during the unrimed snow regime, where mean Doppler velocity is between 1 and $1.5 \, \mathrm{m \, s^{-1}}$ (Fig. 6d). In contrast during the rimed regime very high values of $DWR_{35-95} > 10$ dB around 3 km above ground level correspond to a rapid increase in mean Doppler velocity up to around $2 \, \mathrm{m \, s^{-1}}$. We use this case study to consider the importance of the density factor, particle structure and PSD shape parameter to the triple-frequency signatures of the two regimes (Section 4.1), and evaluate the capability to resolve changes in PSD shape parameter, as well as

the density and internal structure, from triple-frequency radar reflectivity measurements (Section 4.2).

## 4.1 Triple-frequency radar signatures

In Section 3 we explored the influences of various factors on the triple-frequency radar signature, but did not consider the expected ranges of values for each of these parameters, nor how they may co-vary. Here we use in situ and remotely-sensed measurements to explore the size distributions and particle morphologies that best represent the observed triple-frequency radar

signatures.

As a check on the representation of particle properties and radar scattering assumptions, we evaluate the how faithfully our model particles can reproduce the triple-frequency radar signatures measured during the rimed and unrimed snow regimes. We vary the particle internal structure, density and PSD shape parameters based on in situ and remotely-sensed measurements to confirm that the forward model is capable of resolving the triple-frequency radar measurements. The triple-frequency radar

measurements below 2 km in the rimed snow regime are coloured by the average mean Doppler velocity, and contours indicate the 10 th, 50 th and 90 th percentiles of frequency of occurrence (Fig. 7a). The rimed snow is characterised by a flat triple-frequency signature with $DWR_{35-95}$ between 5 and 11 dB and $DWR_{10-35}$ less than 2 dB, and mean Doppler velocities between 1.5 and $2 \, \mathrm{m \, s^{-1}}$. The radar measurements are overlaid with triple-frequency radar signatures for homogeneous spheroids with a range of PSD shape parameters and density factors: the rimed snow exhibits triple-frequency radar signatures, consistent with

homogeneous spheroids with a density factor of $r = 0.5$ (as retrieved from mean Doppler velocity in Mason et al., 2018) and a PSD shape parameter of $\mu = 5$ as measured by PIP at the surface (Fig. 4b). 90% of the triple-frequency radar measurements correspond to median volume diameters between 1.5 and 3 mm, consistent with PIP measurements (Kneifel et al., 2015, and shown later in Fig. 10e.).

The unrimed snow regime (Fig. 7b) exhibits the hook feature characteristic of unrimed aggregates, with most measurements

of $DWR_{35-95}$ between 5 and 10 dB and $DWR_{10-35}$ up to around 10 dB. The most frequent triple-frequency radar measurements are a good fit to aggregates of bullet rosettes with a PSD shape parameter with $\mu = -1$ (measured in situ), and a density factor of $r = 0$. In contrast to the rimed snow, this regime is dominated by lower mean Doppler velocities between 1 and $1.5 \, \mathrm{m \, s^{-1}}$, consistent with the low terminal velocities of even large aggregate snowflakes; less than 10% of the triple-frequency radar measurements include values of $DWR_{35-95}$ greater than 10 dB and mean Doppler velocities greater than $1.5 \, \mathrm{m \, s^{-1}}$, sug-

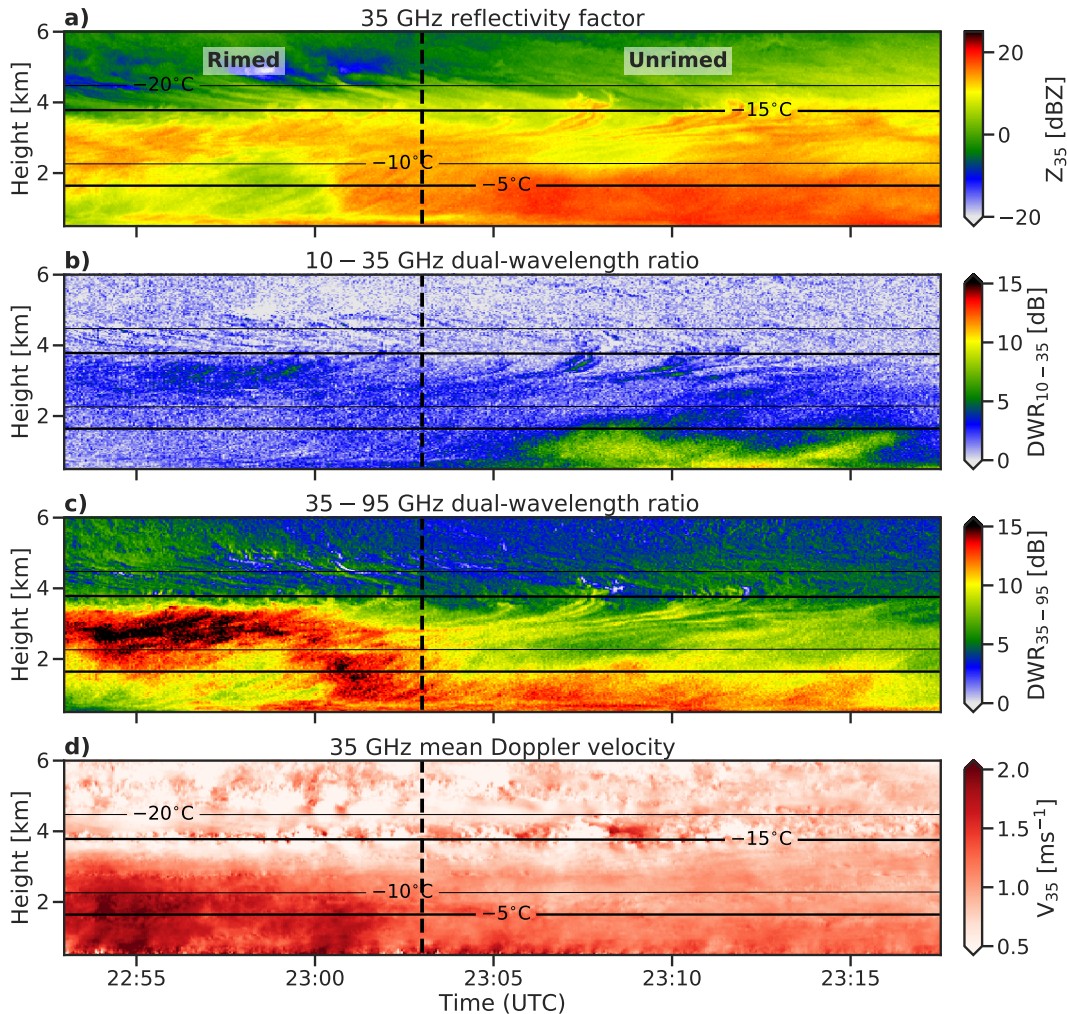

**Figure 6.** Triple-frequency Doppler radar measurements from the February 21 2014 case study at Hyytälä, Finland. (a) 35 GHz radar reflectivity, (b) 10–35 GHz dual wavelength ratio, (c) 35–95 GHz dual wavelength ratio, and (d) 35 GHz mean Doppler velocity. Vertical dashed lines mark the transition between the heavily rimed and large aggregate snowfall regimes; solid lines are contours of temperature from ECMWF analysis.

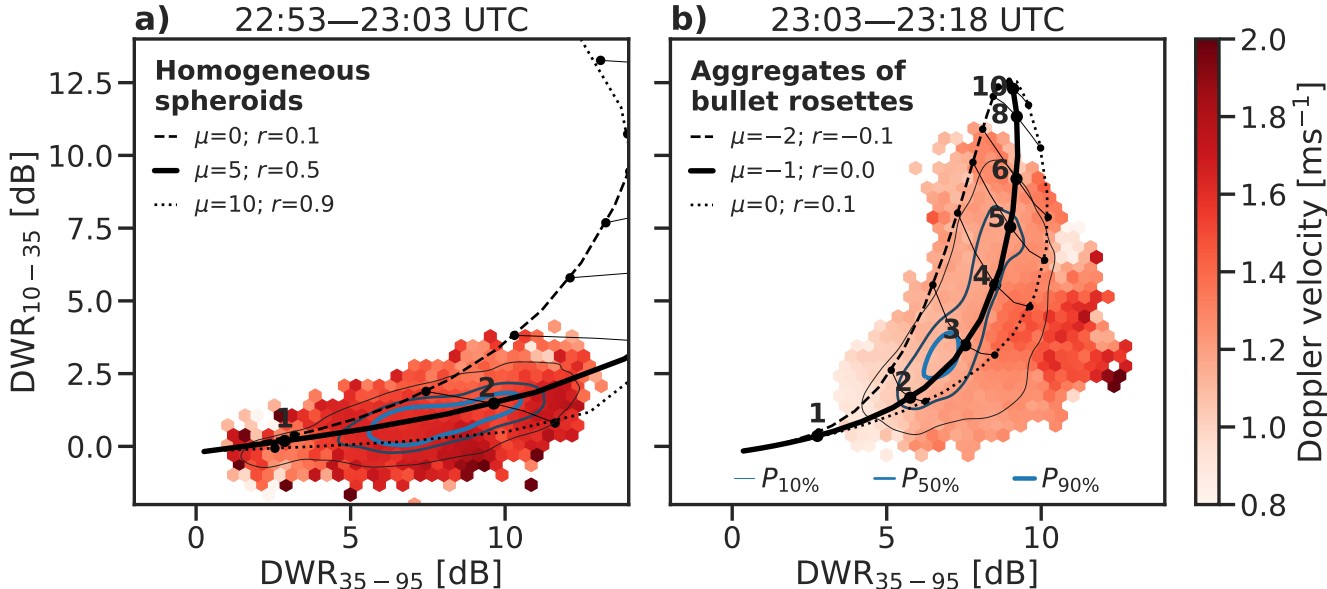

**Figure 7.** Triple-frequency radar measurements below 2 km above ground level in the (a) rimed and (b) unrimed snow regimes of the 21 February 2014 case at Hyytiälä, Finland. Triple-frequency radar measurements are coloured by the corresponding mean Doppler velocity, and frequency of occurrence is indicated with blue contours at the 10th, 50th and 90th percentiles. Overlaid are triple-frequency radar signatures for (a) homogeneous spheroids and (b) fractal particles (aggregates of bullet rosettes; Table 1) with a range of values of PSD shape parameter and density factor that encompass the most frequent triple-frequency radar measurements. Increments of median volume diameter are labelled in millimeters.

gesting a small amount of rimed particles persisting after 23:03 UTC. The triple-frequency radar measurements correspond to fractal particles with median volume diameters from 1.5 mm to 8 mm, but 90% of the data lie within the range expected for fractal particles with median diameters between 2 and 7 mm. This is consistent with in situ PIP measurements at the surface (Fig. 10e).

5      Comparing our particle models against measured triple-frequency radar measurements for a case study, it is evident that the rimed snow is well-represented by a narrow PSD comprising of dense graupel-like particles with a median volume diameter around 1 to 2 mm. The unrimed snow corresponds to a broad distribution of large aggregates with median volume diameters between 3 mm and 8 mm. The fit to triple-frequency radar signatures requires a representation of the PSD shape parameter, particle density and microphysical structure, illustrating that it is necessary to include variability in the PSD shape parameter

10    to resolve the triple-frequency radar signatures of snow.

## 4.2 Triple-frequency radar retrievals

In this section we perform CAPTIVATE retrievals of the 21 February case using triple-frequency and Doppler radar measurements. Mason et al. (2018) used dual-frequency radar reflectivities to constrain two parameters of the PSD, while the density factor—and hence the particle structure—were constrained by mean Doppler velocity at one frequency. Following the studies of Leinonen et al. (2018b) and Tridon et al. (2019), we are interested in whether the triple-frequency radar measurements can be used to constrain a retrieval of particle density, or if—given the evidence presented above—it will instead be necessary to vary the PSD shape parameter in order to satisfy the triple-frequency constraint.

### 4.2.1 Retrievals assuming constant PSD shape parameter

We first perform retrievals assimilating a variety of radar measurements in order to compare their respective contributions. The default retrieval combines radar reflectivity at 10, 35 and 95 GHz and mean Doppler velocity at 35 GHz ($Z_{10,35,95} V_{35}$), which represents the full available measurement vector. As explained in Section 2.2, 95 GHz Doppler velocity measurements are affect by a mispointing; we therefore assimilate mean Doppler velocity from the 35 GHz radar only, and do not consider the contribution of multiple Doppler velocity measurements to this retrieval. To test the capability to retrieve particle density from triple-frequency radar reflectivity factors, we also make a retrieval which does not assimilate Doppler velocity (i.e. $Z_{10,35,95}$); and to test the contribution of the third radar frequency, we test dual-frequency retrievals (i.e. $Z_{10,35} V_{35}$ and $Z_{35,95} V_{35}$). Reducing the number of measurements assimilated may lead to a more challenging inverse retrieval problem, wherein either the measurement vector provides insufficient constraint on an estimate of the state vector, or the state space does not include a solution that satisfies all of the measurements.

The quality of the retrieval is illustrated by comparing the measurements against those forward-modelled from the retrieved state. We take representative profiles from each snow regime: a profile characterised by rimed snow (23:00 UTC; Fig. 8), and one dominated by large unrimed aggregates (23:10 UTC; Fig. 9). The profiles of retrieved variables are shown in the supplementary material.

Comparing retrievals of the rimed profile with an exponential PSD (Fig. 8 a–d), the triple-frequency retrievals $Z_{10,35,95} V_{35}$ and $Z_{10,35,95}$ do not satisfy all three profiles of radar reflectivity simultaneously, with errors of 1 to 2 dB. The dual-frequency radar retrievals $Z_{35,95} V_{35}$ and $Z_{10,35} V_{35}$ satisfy the assimilated radar frequencies, but exhibit large errors in the remaining frequency (e.g. up to 4 dB in forward-modelled $DWR_{10-35}$ for $Z_{35,95} V_{35}$). The profiles forward-modelled by $Z_{10,35,95} V_{35}$ resemble a compromise between the dual-frequency retrievals, but satisfying neither: this is indicative of an over-constrained retrieval, in which the state space does not permit a solution that satisfies all of the observational constraints. The retrievals assimilating Doppler velocity have adequate constraints on the density factor to represent the observed increase in mean Doppler velocity below around 3 km above ground level. $Z_{10,35,95}$ is similar to $Z_{10,35,95} V_{35}$ in terms of radar reflectivity, but the observed profile of mean Doppler velocity, including a rapid increase to around $1.6\,\mathrm{m\,s^{-1}}$ below $2\,\mathrm{km}$ above ground level, is not resolved at all without assimilating mean Doppler velocity.

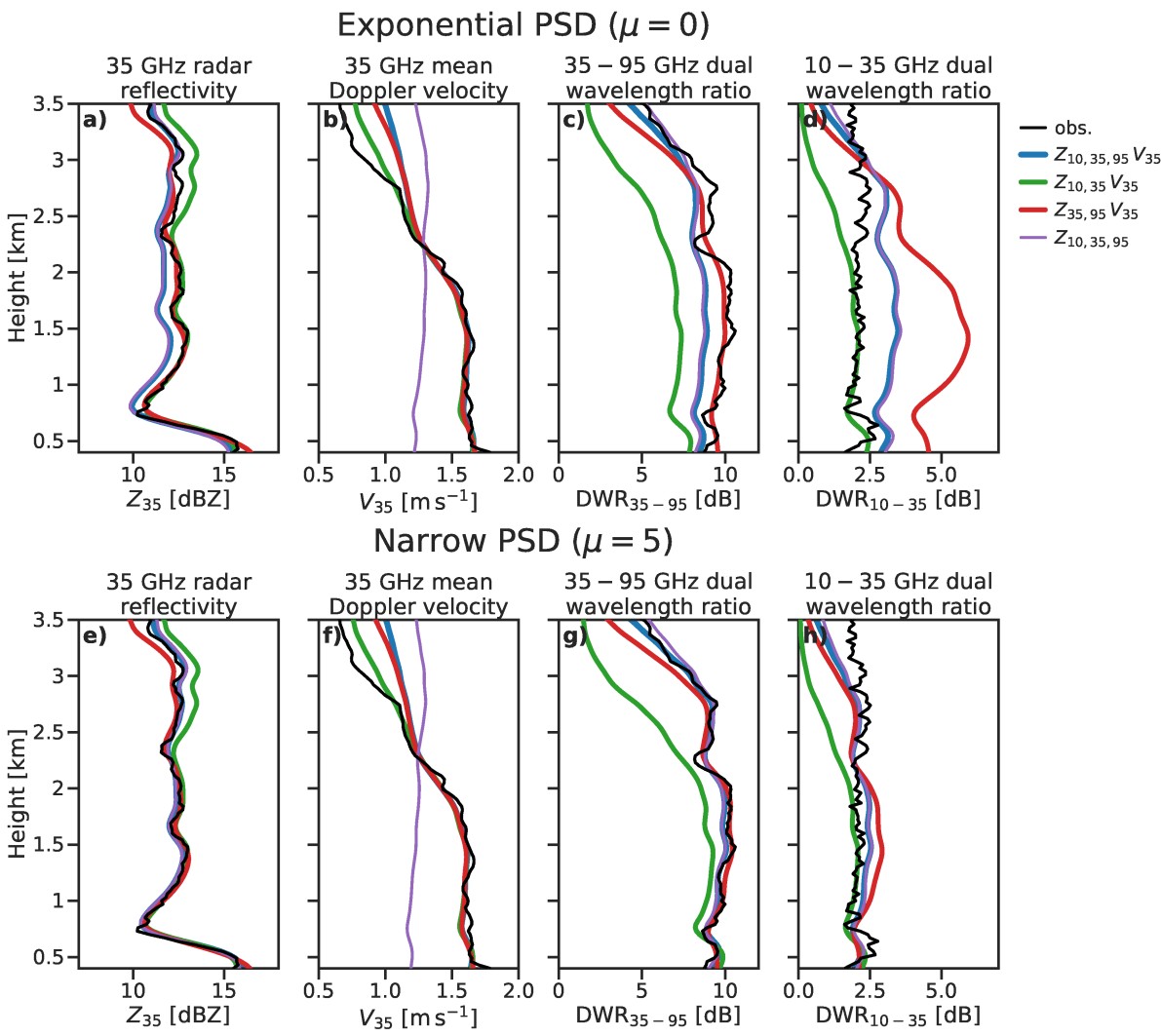

**Figure 8.** Profiles of observed and forward-modelled radar variables for retrievals of a selected profile during the rimed snow regime. Retrievals assuming (a–d) an exponential PSD are compared against those with (e–h) a narrow PSD with $\mu = 5$.

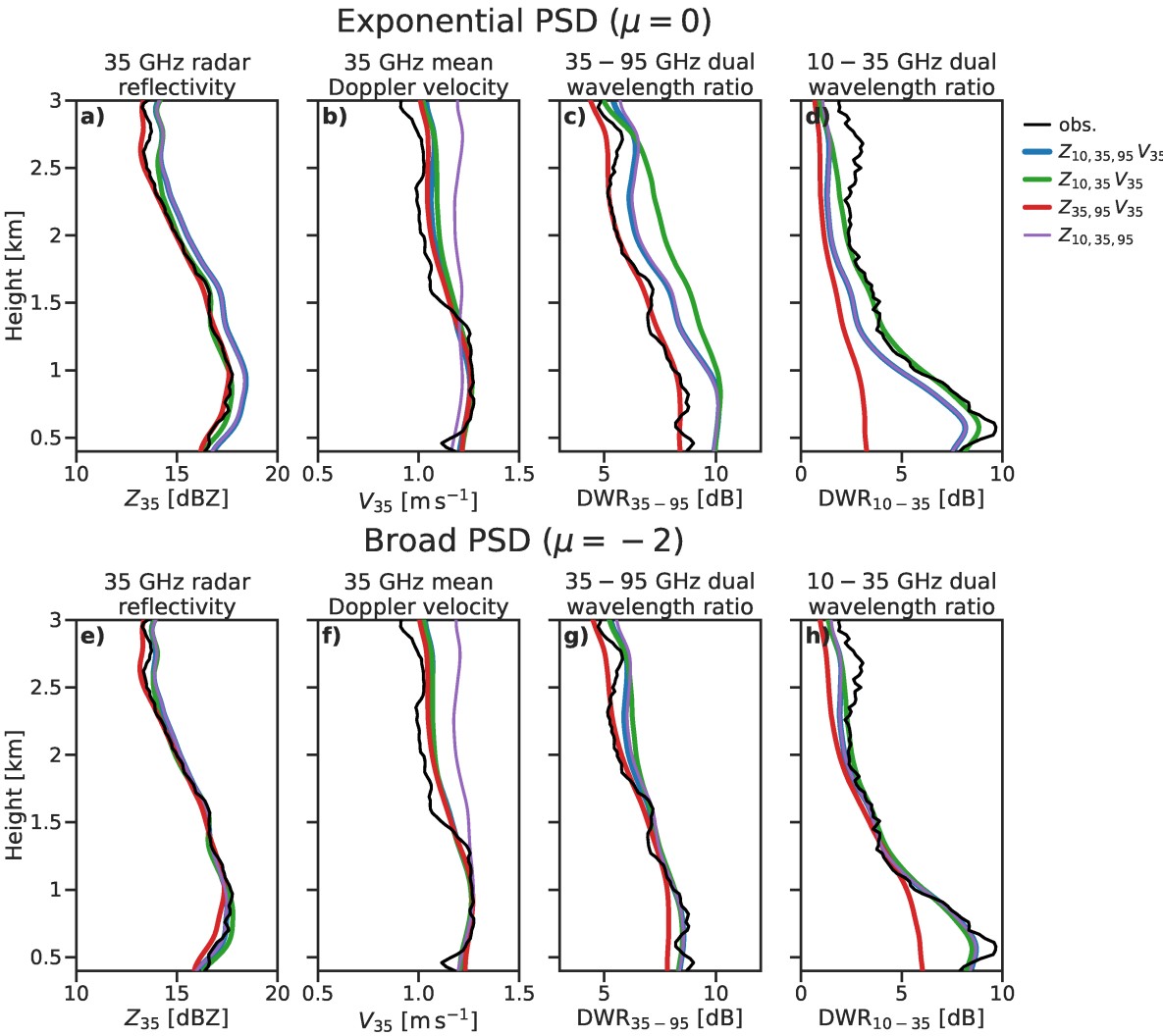

**Figure 9.** Profiles of observed and forward-modelled radar variables for retrievals of a selected profile during the unrimed snow regime. Retrievals assuming (a–b) an exponential PSD are compared against those with (e–h) a broad PSD with $\mu = -2$.

As shown in Section 4.1, the triple-frequency radar measurements in the rimed regime are best explained by a narrow PSD ($\mu > 5$) as indicated by PIP measurements. We therefore repeat the CAPTIVATE retrievals with constant $\mu = 5$ rather than $\mu = 0$ (Fig. 8 e–h). In this retrieval the forward-modelled profiles of radar reflectivity are considerably better constrained by the measurements: $Z_{10,35,95} V_{35}$ especially is very close to all the observations. The exception is that the retrieval does not match the 10 GHz reflectivity factor above around 3 km; it is likely that the PSD shape parameter changes at this level, where the onset of riming is evident from increased mean Doppler velocities, and where layers of supercooled liquid have been identified in the Doppler spectra (Kalesse et al., 2016). The forward-modelled radar reflectivity factors of the dual-frequency retrievals are also better-constrained when a narrower PSD is assumed, indicating that single- and dual-frequency radar retrievals are also affected by the assumption of exponential PSDs. It is not evident that triple-frequency radar measurements provide any significant constraint on the density factor for $Z_{10,35,95}$, as the forward-modelled mean Doppler velocity of that retrieval is unchanged.

Retrievals of the profile of unrimed snow (Fig. 9) show similar results: when an exponential PSD is used, $Z_{10,35,95} V_{35}$ exhibits errors of up to 2 dB in the profiles of radar reflectivity. The retrieval is over-constrained, as the state space does not allow for the triple-frequency radar signature of the large aggregates to be represented. A significant difference between the profiles is that in the unrimed regime retrievals that do not assimilate Doppler velocity are able to represent the observed profile of mean Doppler velocity: this is because the a priori density factor ($r = 0$) makes a reasonable estimate of the terminal fallspeed of unrimed aggregate particles, provided that their size is well-constrained by dual-frequency radar measurements. However, above 1.5 km the non-Doppler retrieval over-estimates the mean Doppler velocity by around $0.2 \, \mathrm{m \, s^{-1}}$, while retrievals that assimilate Doppler velocity estimate negative values of riming factor (Fig.A2d), attributing weaker Doppler velocities above this level to lower-density aggregate snowflakes. As was shown for the rimed regime, all the retrievals are better able to represent the triple-frequency radar measurements when a non-exponential PSD is used: here, assuming a broader PSD ($\mu = -2$) results in consistently better-constrained retrievals at all three radar frequencies. One exception is that the $Z_{35,95} V_{35}$ retrieval is profoundly unable to resolve the very high values of $\mathrm{DWR}_{10-35}$ below about 1 km above ground level. Profiles of retrieved variables (Fig. A2) show that this is due to a failure to resolve the very large aggregates in this part of the profile, to which which the $\mathrm{DWR}_{10-35}$ is most sensitive. This is a result of the hook feature of the triple-frequency radar signature of aggregates, in which $\mathrm{DWR}_{35-95}$ is quickly saturated around 7 to 10 dB. In this situation the third radar frequency provides an important constraint on the size of large aggregate snowflakes.

These results indicate that, at least as the retrieval is configured here, triple-frequency radar measurements are insufficient to constrain a retrieval of the density factor. This is consistent with the results of Leinonen et al. (2018b), wherein the retrieved prefactor of the particle mass-size relation was relatively insensitive to the triple-frequency radar measurements. However, assimilating mean Doppler velocity at one radar frequency provides an effective constraint on the density and structure of snowflakes, as was shown in Mason et al. (2018).

Secondly, we have shown that even when the density factor and particle structure are retrieved, the retrieval can be over-constrained due to the assumption of an exponential PSD. That is, because the PSD shape parameter has a significant influence

on the triple-frequency radar signature (Section 3), it may not be possible to retrieve a state vector that satisfies all three radar frequencies unless the PSD shape parameter can also be estimated, such as from in situ measurements.

### 4.2.2 Retrieval of the PSD shape parameter

We have shown that over-constrained retrievals in both the rimed and unrimed profiles were improved by matching the PSD shape parameter to concurrent PIP measurements at the surface. In CAPTIVATE the PSD shape parameter is assumed constant in each retrieval, but can be configured at runtime. At least for this case study, we have also confirmed that our models for aggregate snowflakes and homogeneous spheroids allow a good representation of triple-frequency radar signatures in conjunction with density and PSD shape parameter (Fig. 7). Since the mean Doppler velocity is used to estimate the density factor and particle structure, it should be possible to minimise errors in the forward-modelled triple-frequency radar measurements, in order to estimate the PSD shape parameter. We carry out a pseudo-retrieval by running multiple $Z_{10,35,95} V_{35}$ retrievals in which the PSD shape parameter takes integer values from $\mu = -2$ to $\mu = 10$. The retrieved value is that which minimises the error in forward-modelled $DWR_{35-95}$ and $DWR_{10-35}$ between 400 and 600 m above ground level (Fig. 10a & b). The forward-modelled DWRs for a range of PSD shape parameters span up to 4 dB, with the range depending on the median volume diameter. To reduce noise in the pseudo-retrieval the minimisation is carried out at a smoothed temporal resolution of 15 s. The estimated PSD shape parameter (Fig. 10 c), and the corresponding retrieved quantities (Fig. 10 d–g), are compared to PIP measurements at the surface.

The retrieved timeseries of the PSD shape parameter is consistent with that measured by PIP, with the rimed regime associated with narrow PSDs ($3 < \mu < 10$) before transitioning to broader PSDs ($-2 < \mu < 0$) in the unrimed regime. Prior to around 23:00 UTC the retrieved PSD shape parameter is especially noisy: high uncertainties reflect the increasingly weak distinctions between forward-modelled DWRs assuming different PSD shape parameters (Fig. 10 a & b). This is because, as observed in Figs. 3 & 4, the triple-frequency radar signatures converge at median volume diameters less than around 2 mm, where non-Rayleigh scattering is weak at all frequencies. As a result, triple-frequency radar signatures using 95, 35 & 10 GHz frequencies provide little insight into ice or snow with median volume diameters below 2 mm. It has been suggested that the inclusion of G-band radars (e.g. 140 or 220 GHz) would provide additional insights into smaller ice particles (Battaglia et al., 2014). Comparison of the retrieval to PSD shape parameters estimated from PIP measurements is especially difficult when the PSD has a median volume diameter below around 1 mm, whereupon disdrometer truncation has a significant effect on estimates of the parameters of the PSD from the method of moments (Moisseev and Chandrasekar, 2007).

The retrievals of snow rate, median volume diameter, number concentration and bulk density at 500 m are all reasonably well-matched to in situ measurements at the surface, given the differences in temporal resolution between the two estimates. The median volume diameter is overestimated by around 50 % in the period of heaviest unrimed snowfall, while retrieved normalized number concentration is within a factor of two of the PIP estimates. While retrieving the PSD shape parameter enables a significant improvement in the representation of triple-frequency radar measurements, we note that it has little impact on the retrieved snow rate or bulk density when compared to a retrieval that assumes an exponential PSD. Compared to a retrieval assuming an exponential PSD, retrieving a broad PSD in the unrimed snowfall results in median volume diameters

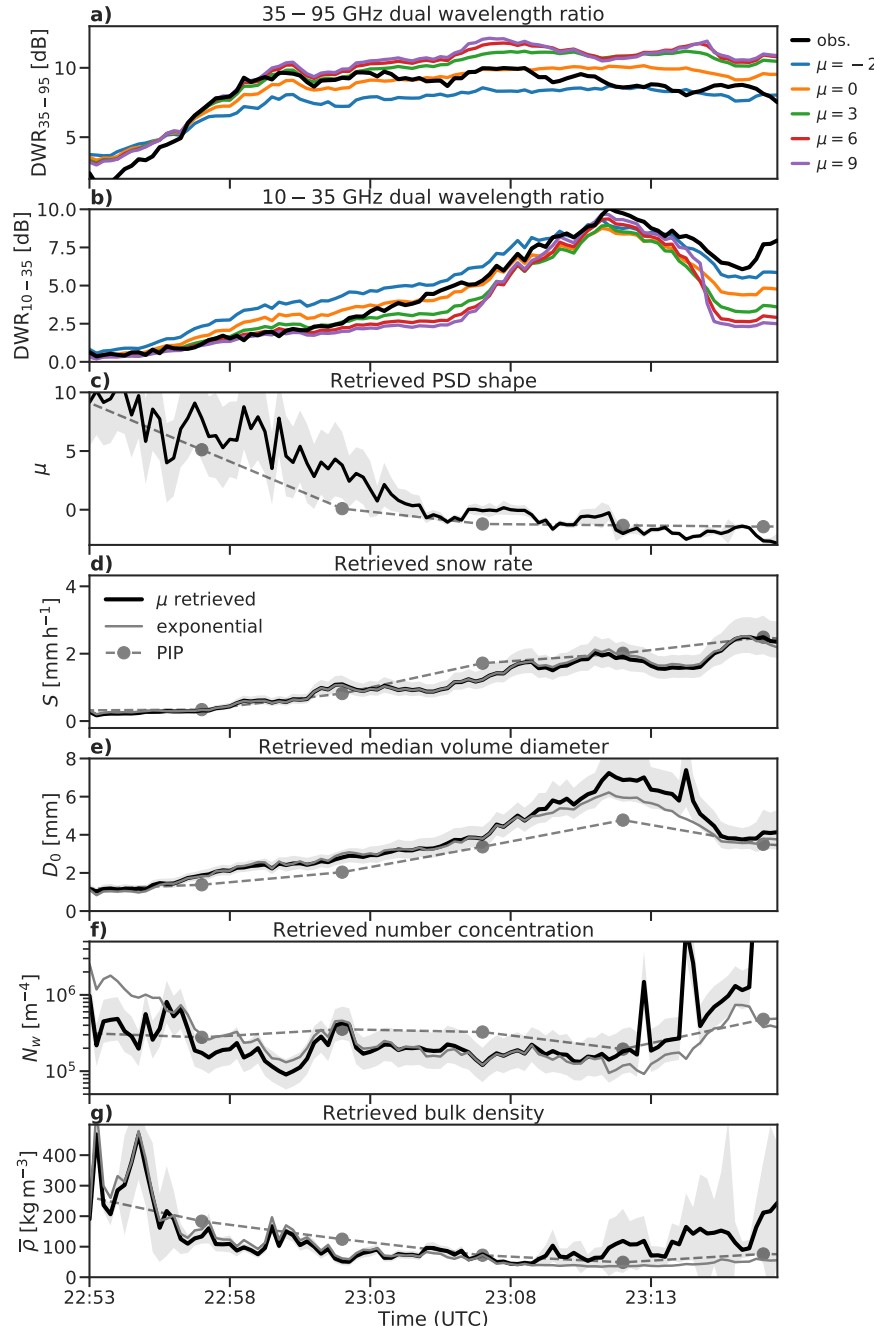

**Figure 10.** Measured and forward-modelled (a) $\mathrm{DWR}_{35-95}$ and (b) $\mathrm{DWR}_{10-35}$ at 500 m above ground level for CAPTIVATE retrievals assuming a range of PSD types; (c) the retrieved PSD shape parameter compared against that estimated from in situ measurements at the surface by PIP; and (d) snow rate, (e) median volume diameter, (f) normalised number concentration, and (g) bulk density comparing values for the pseudo-retrieval against one in which exponential PSDs are assumed. Shaded regions represent the uncertainty in the retrieved values.

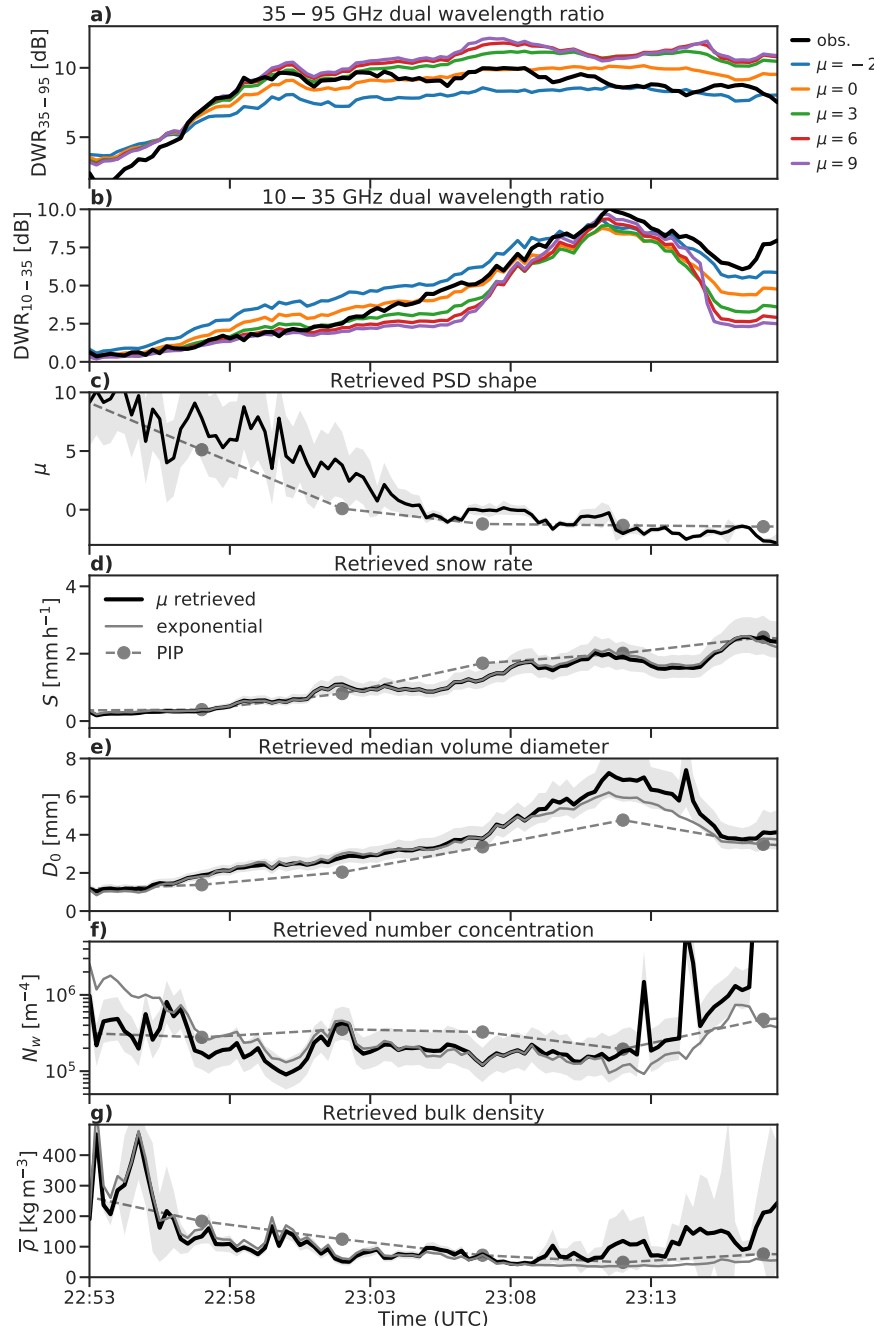

roughly 1 mm larger, while normalised number concentrations are as much as a factor of two lower in the rimed regime and up to a factor of two greater toward the end of the unrimed regime.

## 5 The effect of aggregate internal structure: 16 February 2014 case study

In Section 3 we showed that the triple-frequency radar signature of snow is sensitive to the density factor and internal structure of snowflakes, and also to the shape parameter of the PSD. In Section 4 values of density factor corresponding to riming were constrained by Doppler velocity measurements, while the transition in particle structure between aggregates of bullet rosettes for unrimed snow and homogeneous spheroids for graupel was parameterised by the density factor. This allowed variations in the PSD shape parameter to be constrained from triple-frequency radar reflectivity factors. In this section we consider a case in which the internal structure of unrimed snow varies due to the aggregation of different monomer particles.

Between 00:00 and 01:00 UTC on 16 February 2014 (Fig. 5) a light snowfall between 0.5 and 1.5 m s$^{-1}$ was measured at Hyytiälä (see Section 3.3.1 of Kneifel et al., 2015, for a more detailed discussion of the meteorology). Generating cells are evident in radar reflectivity and DWR above around 4 km or $-20\,°C$; below this level fall streaks indicate that precipitating ice is subject to strong winds down to around 2.5 km or $-8\,°C$. Between $-8$ and $-5\,°C$ rapid increases in radar reflectivity and DWR$_{35-95}$ are evident; this temperature range is conducive to rime splintering (Hallett and Mossop, 1974) and to the growth of needles (Fig. 7 of Kneifel et al., 2015, shows that this temperature range is supersaturated with respect to ice). Kneifel et al. (2015) showed (their Fig. 10) that when the snowfall was dominated by aggregates with very open structures, measured triple-frequency radar signatures exhibited maximum DWR$_{35-95}$ values of around 6 dB, which is significantly weaker than expected for the triple-frequency radar signatures of most aggregates, but consistent with aggregates of large needles (Fig. 3e in Leinonen and Moisseev, 2015, and Fig. 3 herein). Sinclair et al. (2016) used the polarimetric signatures from a nearby scanning radar and a clustering analysis on PIP observations to argue that the snow at the surface comprised a mixture of graupel and small needles—the primary and secondary ice the Hallett-Mossop process—and large aggregates resulting from the rapid growth and aggregation of needles near the surface.

We first explore how the particle models developed in Section 3 can be used to account for the variability in triple-frequency radar signatures in this challenging case (Section 5.1), and we evaluate the effects of variations in ice particle internal structure on CAPTIVATE retrievals (Section 5.2).

### 5.1 Triple-frequency radar signatures

We separate the case into three snow regimes by comparing triple-frequency radar measurements with the signatures forward-modelled for aggregates of needles, aggregates of bullet rosettes, and homogeneous spheroids. In forward-modelling the triple-frequency radar signatures we must also quantify the PSD shape parameter and density factor. PIP measurements indicate a broader than exponential PSD ($\mu = -1$) throughout the case, likely influenced by snowfall comprising distinct populations of large and small particles. Mean Doppler velocities around 1.5 m s$-1$ near the surface (Fig. 5d) indicate the radar measurements

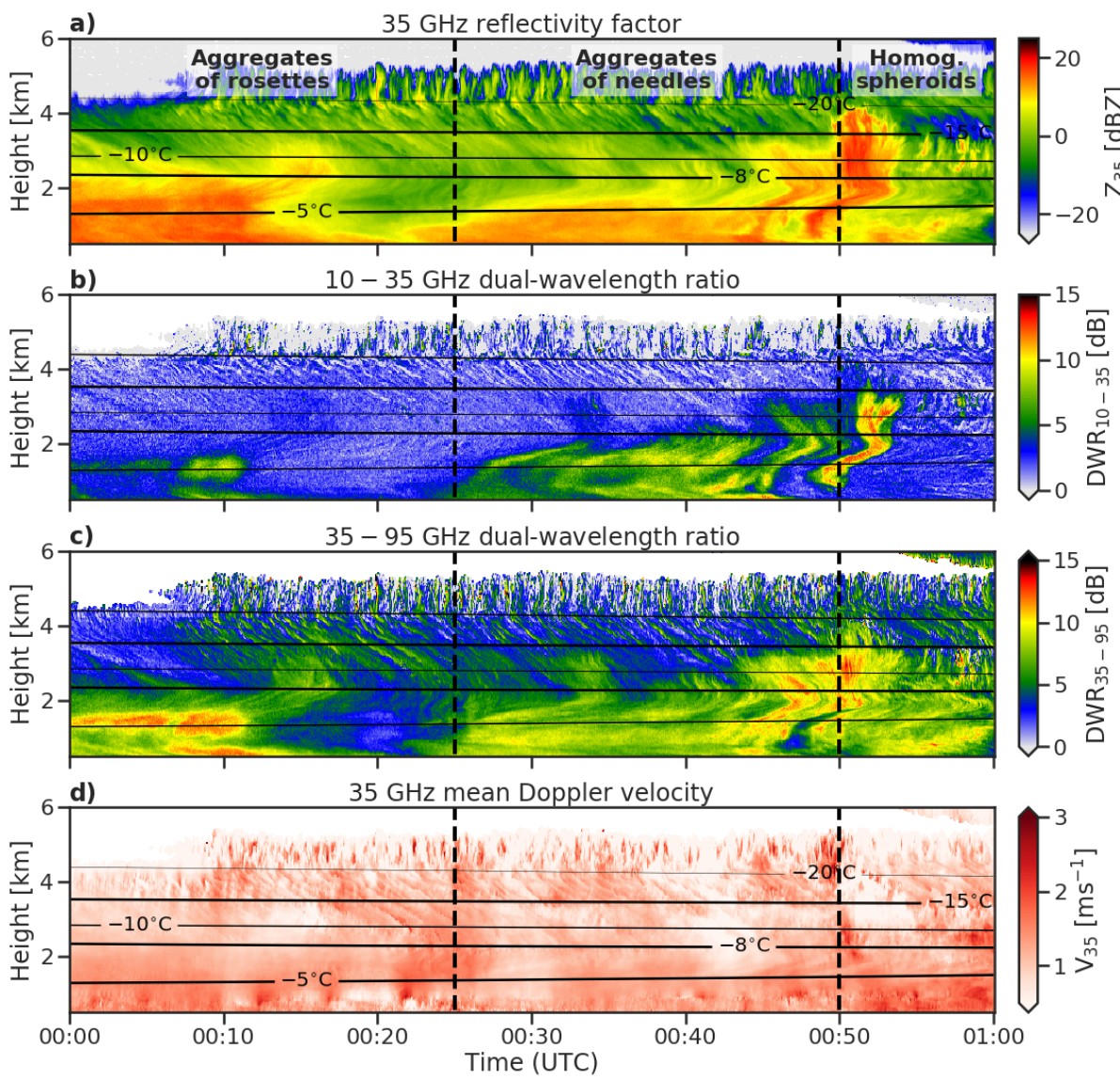

**Figure 11.** Triple-frequency Doppler radar measurements from the February 21 2014 case study at Hyytälä, Finland. (a) 35 GHz radar reflectivity, (b) 10–35 GHz dual wavelength ratio, (c) 35–95 GHz dual wavelength ratio, and (d) 35 GHz mean Doppler velocity. Vertical dashed lines mark the transition between the rimed and unrimed snowfall regimes; solid lines are contours of temperature from ECMWF analysis.

are dominated by unrimed particles ($r \approx 0$), except toward the end of the case where mean Doppler velocities approach $2\,\mathrm{m\,s^{-1}}$, and more graupel is observed.

Triple-frequency radar measurements during the first regime (00:00 to 00:25 UTC; Fig. 12a) are most consistent with aggregates of bullet rosettes. Median volume diameters are up to 5 mm, with most between 1.5 and 3.5 mm; at the surface, PIP measures $D_0 = 2.5\,\mathrm{mm}$ (Fig 13c). The second regime (00:25 to 00:50 UTC; Fig. 12b) is characterised by the lower values of $\mathrm{DWR}_{35,95}$ associated with aggregates of needles. The triple-frequency measurements suggest median volume diameters up to 1 cm, with most between 3 and 4.5 mm, which is slightly higher than measured by PIP. The spread of the measurements lateral to the forward-modelled triple-frequency radar signatures may be indicative of a particles with a range of internal structures within this regime, such as may result from aggregates from different sized needles. Toward the end of the snowfall event (00:50 to 01:00 UTC; Fig. 12 c) the flatter triple-frequency radar signature is consistent with small homogeneous spheroids with median volume diameters less than 2 mm. Relatively large density factors ($r > 0.5$) are required to match the dominant flat triple-frequency signature for such small particles. Some measurements with $\mathrm{DWR}_{10-35} > 3$ dB are a reminder that this regime includes a mixture of particle types, the triple-frequency radar signatures of which are consistent with a small amount of the aggregates of needles.

This indicates that the full range of triple-frequency radar measurements cannot be covered by transitioning between a single model for aggregate particles and homogeneous spheroids. The distinct internal structure of some particles, such as aggregates of needles, must also be represented. While these regimes and corresponding particle models are identified based on the measured triple-frequency signatures, we note that the snow falling throughout this event comprises a mixture of these particle types (Sinclair et al., 2016). In the particle imagery shown in Kneifel et al. (2015) the largest aggregates of needles were sampled from 00:38 to 00:50 UTC, but graupel was also observed. Prior to 00:25 UTC the particle imagery shows both graupel and needles inter-mixed with large aggregates; finally, after 00:50 UTC, aggregates of needles are observed as well as graupel.

## 5.2 Triple frequency radar retrievals

We will now evaluate the capabilities and limitations of CAPTIVATE retrievals assuming a range of particle models. Unlike the February 21 case, PIP measurements indicate a broad PSD ($\mu = -1$) throughout this case; we therefore set $\mu = -1$ for all retrievals, acknowledging that this prior knowledge from in situ measurements is not always available for remotely-sensed estimates of snow. To consider the importance of resolving different particle structures, we carry out three retrievals, using each of the particle types considered in Section 3: aggregates of needles, aggregate of bullet rosettes, and homogeneous spheroids. In order to expand the range of triple-frequency radar signatures represented, an additional distinction is made between aggregates of "large" and "small" needles, where the latter corresponds to the fit to aggregates of needles in Section 3, and the former takes a lower value of power-law exponent ($\gamma = 4/3$ instead of $\gamma = 5/3$); this is consistent with the relation of the triple-frequency radar signature to larger needle monomers observed by Leinonen and Moisseev (2015). Unlike in the previous case, the hybrid approach transitioning between aggregates of bullet rosettes at low density factors and homogeneous spheroids at high density factors is not used. As such, we cannot expect any one retrieval to accurately represent the entire case, and it may be that we

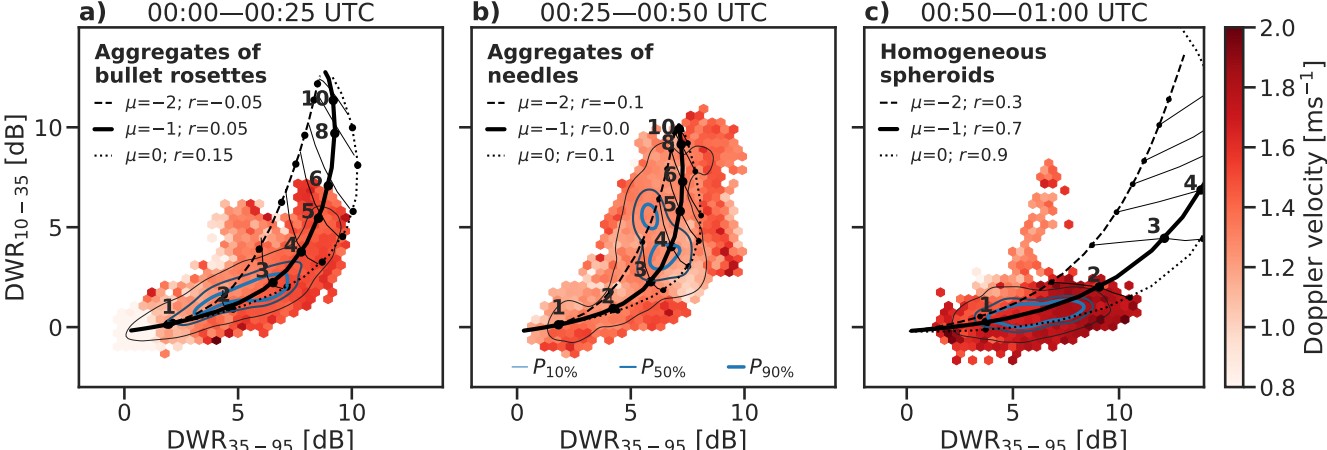

**Figure 12.** Triple frequency radar measurements from below 1 km above ground level during the 16 February case. The data are divided into three regimes and overlaid by a range of particle models, as informed by PIP measurements. Throughout this case a PSD shape parameter of $\mu = -1$ was measured in situ. (a) The period 00:00–00:25 UTC is consistent with the triple-frequency radar signature of aggregates of bullet rosettes (Table 1), (b) the period 00:25–00:50 UTC is to aggregates of needles (Table 1), and (c) the period 00:50–01:00 UTC to homogeneous spheroids. The triple-frequency radar measurements are coloured by the average mean Doppler velocity, and blue contours mark the 10th, 50th and 90th percentiles of the frequency of the data.

are unable to adequately resolve the features of snow known to comprise a mixture of different particle types. Nevertheless, by comparing these retrievals we hope to show how effectively variations in the internal structure of ice particles may improve the fit to triple-frequency radar measurements, and also how sensitive the retrieved quantities are to different particle models.

The forward-modelled dual-frequency ratios (Fig. 13a & b) show how well-constrained each retrieval is by the observed variables. PIP measurements of the PSD shape parameter and the constant assumption of $\mu = -1$ are compared in Fig. 13c, and Fig. 13d–g show the retrieved estimates and PIP measurements of snow rate, median volume diameter, normalized number concentration and bulk density.

The spread in forward-modelled DWRs are as much as 3 dB when the measured DWRs are large; however, when the measured DWR is low (i.e. toward the lower-left part of the triple-frequency radar signature diagram) the retrievals are almost insensitive to the particle model. The maximum values of $\mathrm{DWR}_{35-95}$ (Fig. 13a) are indicative of the maximum extent of the hook feature of the triple-frequency diagram: accordingly, the aggregates of needles have the lowest maximum values, saturating around 6 or 7 dB. Conversely, the lowest values of $\mathrm{DWR}_{10-35}$ (Fig. 13) correspond to homogeneous spheroids, while the greatest are for the aggregates of needles. The retrieved snow rate is sensitive to particle structure with differences of 0.5 to 1 mm h$^{-1}$ between aggregates of large needles and homogeneous spheroids, except where the measured DWR are lowest. The estimates of median volume diameter and normalized number concentration assuming homogeneous spheroids are closer to the PIP measurements than the estimates assuming aggregate snowflakes. However, when homogeneous spheroids are assumed the bulk density is overestimated by around 100 % with respect to PIP measurements.

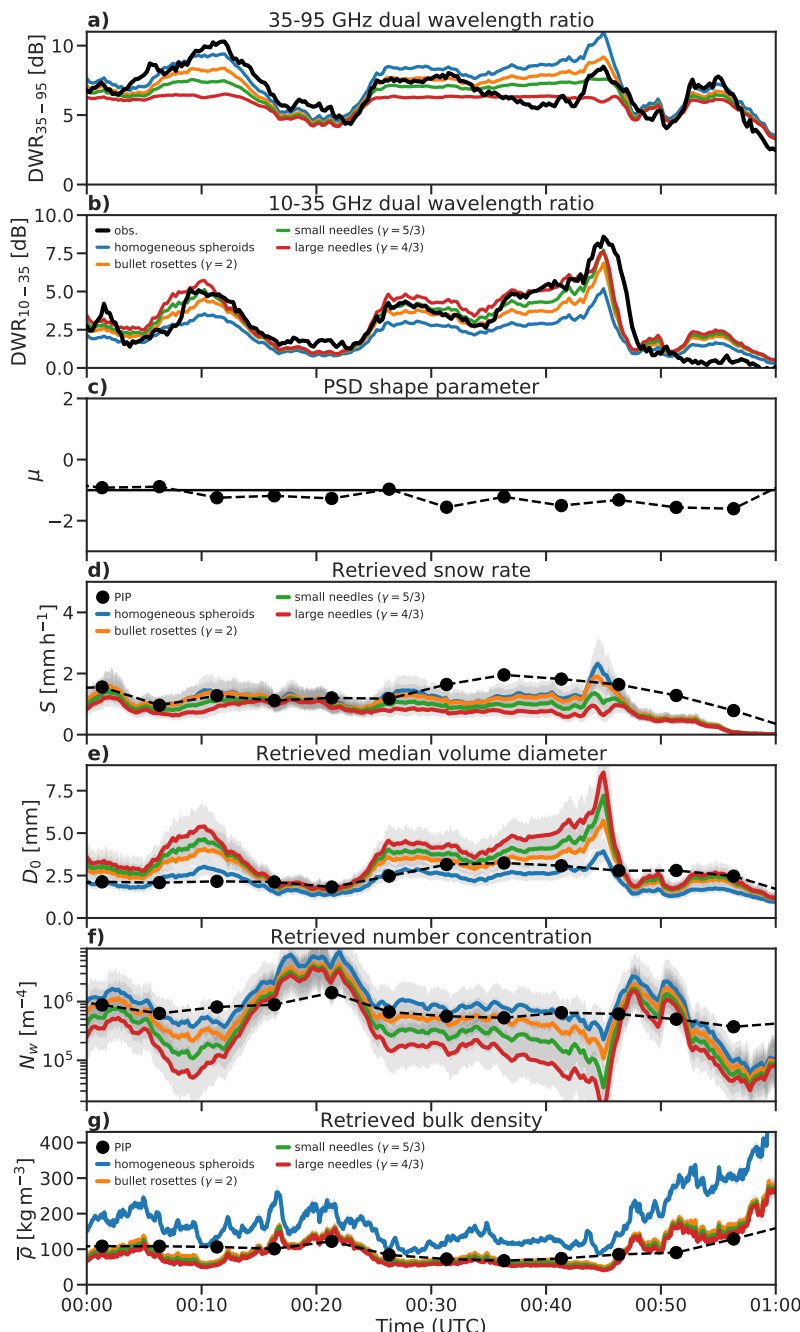

**Figure 13.** Triple-frequency Doppler radar retrievals from the 16 February 2014 case during BAECC, assuming homogeneous spheroids, aggregates of bullet rosettes, and aggregates of large and small needles. The forward-modelled 35-94 GHZ (a) and 10-35 GHx (b) dual-frequency ratios are compared against radar measurements. The assumed PSD shape parameter $\mu = 1$ for all the retrievals reflects the near-constant in situ measurements throughout the case (c). The retrieved snow rate (d), median volume diameter (e), normalized number concentration (f) and bulk density (g) at 500 m above ground level are compared against PIP measurements at the surface.

While the triple-frequency radar signatures of the three regimes were broadly represented by three different particle types, the time series shows much more variability. In the first regime (00:00–00:25 UTC), the triple-frequency radar signature of which was a fit to aggregates of bullet rosettes, $DWR_{35-95}$ takes large values associated with graupel, while $DWR_{10-35}$ jumps between values consistent with graupel and the larger ratios more characteristic of aggregates. These fluctuations are likely due

to the mixture of different particle types. The second regime (00:25–00:50 UTC) is a more consistent fit to aggregates of needles: between 00:25 to 00:35 UTC the measurements correspond to aggregates of small needles, while between 00:35 to 00:45 UTC—where the lowest values of $DWR_{35-95} \approx 6\,dB$ are observed—the aggregates of larger needles offer a good fit to measurements. While the aggregates of needles dominate the triple-frequency radar signature, this regime is associated with the greatest errors in retrieved snow rate, which is roughly half that of the PIP retrieval, in conjunction with lower normalized

number concentrations and much larger median volume diameters. It seems likely that, due to CAPTIVATE assuming a single population of particles and being constrained by the largest particles, the contribution of graupel to the snow rate is being disregarded. Finally, after 00:50 UTC $DWR_{10-35}$ decreases to less than 3 dB, consistent with compact graupel-like particles. The differences between retrievals are relatively small in this regime: all tend to underestimate the snow rate compared to PIP, but correctly resolve the increase in bulk density.

The range of forward-modelled DWRs for retrievals assuming particles with different internal structures are as much as 3 dB. As observed in the triple-frequency radar signatures, this is of a similar magnitude to the variations due to uncertainties in the density and PSD shape. Unlike the PSD shape parameter, uncertainty in particle structure results in a spread in retrieved snow rates. This case includes both rime splintering and rapid aggregation of secondary ice particles, leading to snowfall at the surface comprising compact graupel, needles, and large aggregates of needles. Secondary ice poses a challenge for radar

retrievals. The triple-frequency radar signatures are dominated by the largest particles, which are slow-falling aggregates. Assuming this particle type within the retrieval neglects compact and fast-falling graupel, leading to an underestimate of the snowfall at the surface. Conversely, assuming homogeneous spheroids for the retrieval results in a better fit to the snow rate, but the retrieval significantly over-estimates bulk density and is poorly constrained by the triple-frequency radar reflectivity factors.

**6   Discussion and conclusions**

It has been well-established in theoretical and observational studies that triple-frequency radar measurements of snow can provide insights into the microphysical structure and density of ice particles (Kneifel et al., 2011; Leinonen et al., 2012; Kneifel et al., 2015; Stein et al., 2015). In the context of recent ground-based (e.g. Petäjä et al., 2016) and airborne (e.g. Houze et al., 2017) multiple-frequency radar measurements, large databases of realistic ice particles from numerical simulations and

in situ measurements are being used to better understand the links between ice microphysics and multiple-frequency radar scattering properties (Kneifel et al., 2018). One important application of this research is to assimilate triple-frequency radar measurements in radar retrievals that estimate variations in ice particle properties, such as due to riming (Leinonen et al., 2018b; Tridon et al., 2019). A significant challenge in developing this capability is to sufficiently represent the parameters that

influence the triple-frequency radar signature in order to either estimate them within retrievals, or to quantify their contributions to retrieval uncertainties.

We began this study by using approximations to radar backscatter cross-section of ice particles to explore the parameters that most strongly affect their triple-frequency radar signatures. The triple-frequency radar signature varies due to both the density and homogeneity of ice particles, consistent with the conceptual model of Kneifel et al. (2015). For aggregate particles represented by the self-similar Rayleigh-Gans approximation (SSRGA; Hogan et al., 2017), the magnitude and scale-dependence of random fluctuations in internal structure are also important, as explored by Stein et al. (2015) and Leinonen and Moisseev (2015). The SSRGA provides physically meaningful parameters by which both the shape of the average particle, and the small-scale fluctuations in the structure represented by a power spectrum, can be tuned. To inform our interpretation, we fit SSRGA coefficients to simulated aggregates of a range of monomer types, and showed that aggregates of needles have a distinct triple-frequency radar signature from aggregates of other particles (see also Leinonen and Moisseev, 2015). Finally, while most radar studies of snow assume exponential PSDs, we showed that the PSD shape parameter has a similar effect on triple-frequency radar measurements to particle density by modifying the relative importance of features in the radar backscatter spectrum. Next, we used radar and in situ measurements of snow to disentangle the multiple parameters affecting observed triple-frequency radar signatures.

We evaluated our particle models against triple-frequency radar measurements and particle properties from in situ observations of two snow events from the BAECC 2014 field campaign at Hyytiälä, Finland (Kneifel et al., 2015). First we applied information from particle imaging measurements to select the particle model that best represents the triple-frequency radar signature for each snow regime. In a case including heavily rimed graupel and unrimed aggregates, the triple-frequency radar signature of graupel was consistent with homogeneous spheroids with a moderate density factor and a narrow PSD, while the large unrimed aggregates were represented by aggregates of bullet rosettes with a broad PSD. In a contrasting case including the production and rapid aggregation of secondary ice particles due to rime splintering, triple-frequency signatures were indicative of a mixture of unrimed aggregates of bullet rosettes and aggregates of needles, and of heavily rimed compact graupel particles. Based on in situ measurements during this case, significant fluctuations in the triple-frequency radar signature can be confidently attributed to the internal structure of the aggregates, rather than to changes in the PSD shape or particle density.

Based on these insights, we sought to better understand how the density and internal structure of particles, and the PSD shape parameter, can be included or accounted for in triple-frequency radar retrievals. We ran a range of CAPTIVATE (Mason et al., 2018) retrievals for each case, and compared remotely-sensed estimates to PIP measurements at the surface. In the case including graupel and unrimed aggregates, we demonstrated that the assumption of an exponential PSD resulted in triple-frequency radar retrievals that were over-constrained. Only when non-exponential PSDs were assumed, informed by PIP measurements, was the retrieval able to satisfy all three radar reflectivity factors. Further, we found that while the mean Doppler velocity can be exploited to estimate both the density factor and the transition between aggregate snowflakes and homogeneous graupel particles (as in Mason et al., 2018), the density factor was not well-constrained by triple-frequency radar reflectivity factors. This is consistent with Leinonen et al. (2018b), who found the prefactor of the particle mass-size relation was insensitive to the assimilation of a third radar frequency. However, the third radar frequency proved essential to constraining the median volume

diameter of aggregate snowflakes: due to the characteristic hook feature of their triple-frequency radar signature, $\text{DWR}_{35-95}$ becomes saturated at values around 7 to 10 dB, and provides little information on the size of large aggregates. Next we demonstrated a novel approach to use the third radar reflectivity factor to estimate the PSD shape parameter. This pseudo-retrieval was made by minimising errors in DWRs from an ensemble of retrievals assuming a range of PSD shape parameters, and the result compared well with PIP measurements at the surface. Accounting for variations in the PSD shape parameter is necessary to assimilate triple-frequency radar reflectivity factors, and retrievals of the PSD shape parameter may provide insights into microphysical processes in snow. In the case of heavy riming studied here, narrow PSDs were strongly correlated with riming and broad PSDs with aggregation; while this is consistent with some other observations (Garrett et al., 2015), we note that Tiira et al. (2016) found only a weak relation between bulk ice density and the PSD shape parameter measured during BAECC, and we anticipate further research associating PSD shape with aggregation and riming processes in snow. An important result, however, is that varying the PSD shape parameter does not significantly change the retrieved snow rate, justifying the common assumption of exponential PSDs for single- and dual-frequency radar retrievals (Delanoë et al., 2005). Nevertheless, the capability to remotely-sense changes in the PSD shape parameter alongside other properties of snow may provide significant insights into processes of aggregation and riming in snow.

Retrievals of the second case study focused on the importance of the internal structure of aggregate snowflakes. In this case the PSD shape parameter at the surface was near-constant. While the presence of graupel was evident in the particle imaging measurements and is necessary for the rime splintering process, the triple-frequency radar signatures were dominated by large aggregates. CAPTIVATE retrievals of this case assumed a range of different particle structures from homogeneous spheroids to aggregates of large needles. In the regime where the largest aggregates of needles were observed at the surface, the retrieval assuming these particles was best able to represent the assimilated triple-frequency radar measurements; however, this corresponded to an significant underestimate of surface snow rate. Conversely, assuming graupel-like particles resulted in a weak fit to the assimilated measurements but a better representation of surface snowfall. This illustrates the challenges of secondary ice production to radar retrievals, wherein a single particle population is typically assumed: because the radar measurements are dominated by the largest aggregates, the contribution of dense, fast-falling graupel particles to the snowfall is neglected in the retrieval. Unlike the PSD shape parameter, variations in particle internal structure contribute significant uncertainties in the retrieved snow rate; however, we note that aggregates of most monomers appear to exhibit broadly similar triple-frequency radar signatures, to which aggregates of large needles are an exception. It is not clear how frequently biases in radar retrievals due to these effects may be expected to affect snowfall estimates; however, the ranges of temperatures at which these occur are well understood (e.g. Korolev et al., 2000; Heymsfield et al., 2013).

We have identified a number of parameters that affect the triple-frequency radar signature of snow and should therefore be considered within triple-frequency radar retrievals. Two short case studies have been used to demonstrate contrasting situations in which the effects of these parameters, as represented within models for the size distribution, morphology and radar backscatter cross-section of ice particles, can be observed in radar measurements and affect radar retrievals of snowfall. Particle density and the transition from aggregate snowflakes to more homogeneous graupel particles, which are inter-related in the riming process, have dominated the conceptual model explaining the triple-frequency radar signature (Kneifel et al., 2015),

and were included in the retrieval of Mason et al. (2018). The PSD shape parameter has usually been excluded from studies of the triple-frequency radar signature by assuming an exponential PSD, an assumption which is justified in radar retrievals in many situations by the weak effect of the PSD shape parameter on the snow rate. However, the influence of the PSD on the triple-frequency radar signature is similar in magnitude and sense to that of the density factor, and misinterpretation of

triple-frequency radar data may result if both parameters are not accounted for. In a recent study, Grecu et al. (2018) demonstrated retrievals from airborne triple-frequency radars informed by PSDs measured in situ, rather than assuming parameterised PSDs. Consistent with the present study, they found that including variability in particle spectra resulted in greater ambiguity between the triple-frequency radar signatures of different particle types than may be expected from studies assuming exponential PSDs for all particles (e.g. Kneifel et al., 2011; Leinonen and Szyrmer, 2015). While the density factor and the transition

from aggregates to homogeneous spheroids are represented, the PSD shape parameter and variations in the internal structure of aggregate snowflakes are not currently implemented as state variables within CAPTIVATE. Including such parameters in the radar forward model would allow for information about their natural variability to be propagated into the uncertainties of retrieved quantities. If adequately constrained by radar and synergistic measurements, some parameters such as the PSD shape parameter may be retrieved. The challenges of quantifying both secondary ice and variations in the PSD may be best addressed

using Doppler spectra (e.g. Kneifel et al., 2016), and this should be the subject of future work.

Two case studies from BAECC 2014 at Hyytiïä, Finland, have been used to demonstrated the impact of particle size distribution and internal structure to triple-frequency radar measurements, but further work is needed to establish the relative importance of variations in these parameters to estimates of global snowfall. The evaluation of triple-frequency Doppler radar retrievals against in situ particle properties depends on high-quality remotely-sensed measurements supported by in situ ob-

servations at the surface (Tiira et al., 2016; von Lerber et al., 2017; Moisseev et al., 2017) in a location where wintertime precipitation is frequently not affected by melting. The significant challenges of colocating and cross-calibrating multiple radars and correcting for attenuation due to supercooled liquid water—any of which are potential biases and uncertainties in the present retrieval—have been addressed by Kneifel et al. (2015) and Dias Neto et al. (2019). The general scarcity of such high quality datasets and their importance for evaluating our models of ice particle morphology and size distribution, and

for testing retrievals against in situ measurements, make the case for further and ongoing deployments of multiple-frequency Doppler radar instruments at a range of ARM and Cloudnet field sites. Increasing the geographic diversity of ground-based snow studies, as well as the quantity of measurements, will be critical to increasing confidence in the use of models of ice particles and microphysical processes for global snowfall estimates.

In addition to developing the capability for more advanced radar retrievals, insights into ice and mixed-phase microphysics

from multiple-frequency and Doppler radar measurements of snow should be focused on informing the ice particle models used in other radar retrievals, and especially snowfall estimates from spaceborne radars such as the 94 GHz Doppler cloud profile radar aboard EarthCARE (Illingworth et al., 2015). The present work contributes to a more detailed understanding of the uncertainties of radar retrievals due to variations in the properties of snow. In planning for satellite missions beyond EarthCARE (National Academies of Sciences Engineering and Medicine, 2018), the benefits of dual-frequency radar as a

constraint on the size and number concentration of hydrometeors has been well-established; here, we have demonstrated that

triple-frequency radar measurements may be used to constrain additional properties of the morphology and size distribution of snow, to provide insights into ice and mixed-phase cloud microphysics.

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

## Appendix A: Retrieved profiles

The profiles of retrieved variables corresponding to the retrievals described in Section 4.2 are Fig. A1 for the unrimed regime 20 and Fig. A2 for the rimed regime.

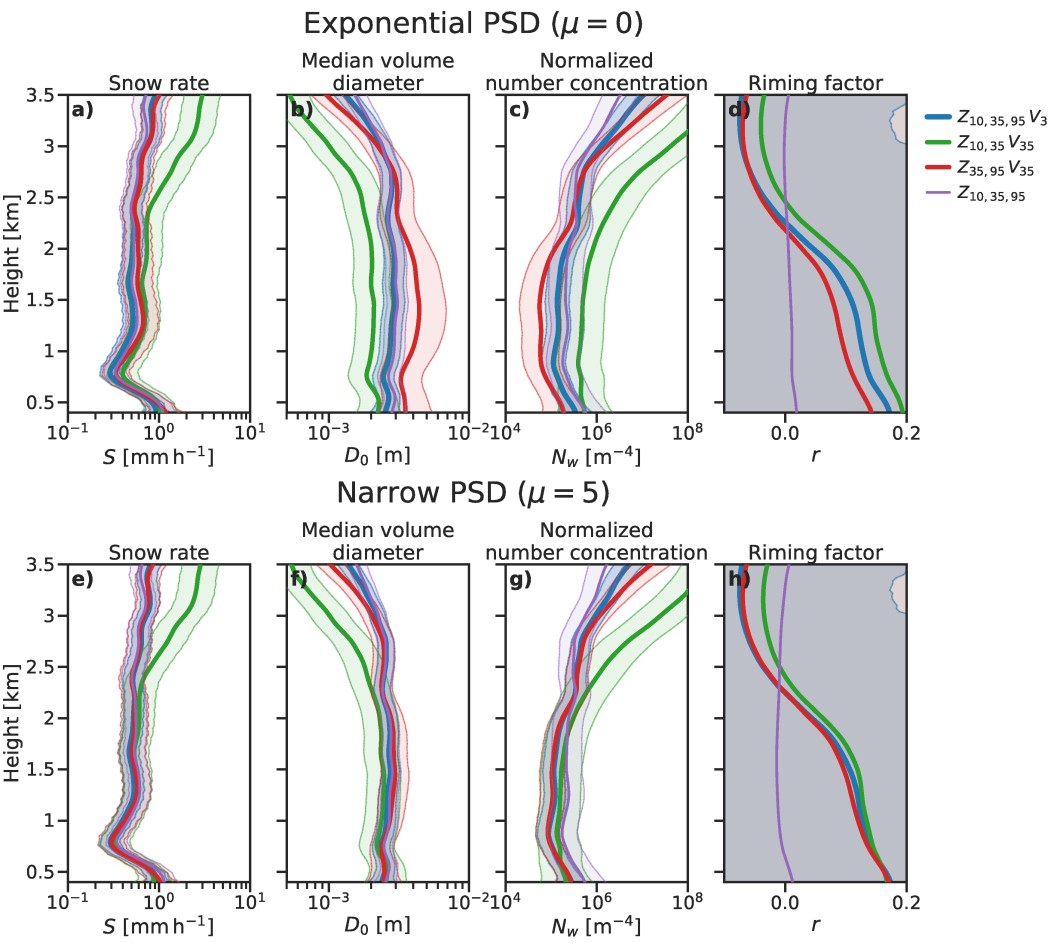

**Figure A1.** Profiles of retrieved variables for retrievals of a selected profile during the rimed snow regime. Retrievals assuming (a–d) an exponential PSD are compared against those with (e–h) a narrow PSD with $\mu = 5$.

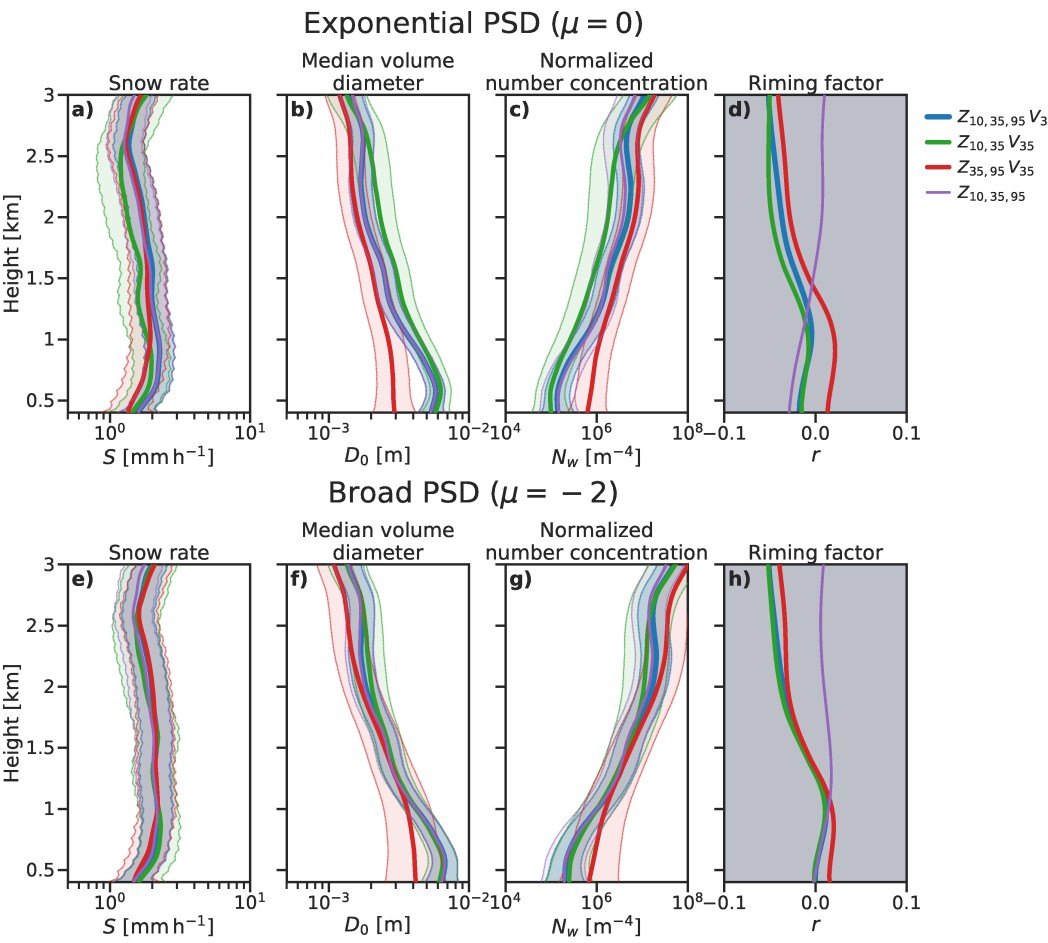

**Figure A2.** Profiles of retrieved variables for retrievals of a selected profile during the unrimed snow regime. Retrievals assuming (a–b) an exponential PSD are compared against those with (e–h) a broad PSD with $\mu = -2$.

*Author contributions.*

*Competing interests.* The authors do not have any competing interests.

*Acknowledgements.* This work is supported by the National Centre for Earth Observation (NCEO) and European Space Agency Grant 4000112030/15/NL/CT, with computing resources provided by the University of Reading. D. Moisseev acknowledges funding from ERA-PLANET, transnational project iCUPE (Grant Agreement 689443), funded under the EU Horizon 2020 Framework Programme and Academy of Finland (grants 307331 and 305175). Contributions by S. Kneifel were carried out within the Emmy-Noether Group OPTIMIce funded by the German Science Foundation (DFG) under Grant KN 1112/2-1.