# Peer review of "The importance of particle size distribution and internal structure for triple-frequency radar retrievals of the morphology of snow"

_Atmospheric Measurement Techniques, 2019_

## Referee Comment (RC1) · Anonymous Referee #1 · 19 Apr 2019

Title: The importance of particle size distribution shape for triple-frequency radar retrievals of the morphology of snow

Authors: Mason, Shannon L. Hogan, Robin J. Westbrook, Christopher Kneifel, Stefan Moisseev, Dmitri

DOI: amt-2019-100

General Comments: Overall, this is a very well-written and insightful manuscript. It presents novel results looking at the efficacy of triple-frequency retrievals while still building off prior results, putting its results in context of recent literature, and suggesting pathways for future research. I found no major concerns with any of the underlying

science; the majority of my comments are minor and related to improving the clarity of the manuscript. For the most part the figures are very clear while containing a lot of information for readers. Based on this, I recommend the paper for publication pending the below corrections.

Specific Comments:

1. This may be personal preference, but throughout the paper "PSD shape" is used to refer to the parameter $\mu$ – while I understand in the normalized distribution space $\mu$ does modify the actual shape (width) of the distribution, when talking about $\mu$ I think it may be clearer to refer to this as the "PSD shape factor" or "PSD shape parameter", as done on P4 L9.

2. P2 L30-L32: Are triple-frequency measurements always at precisely 95 GHz, 35 GHz, and a third frequency below 15 GHz? Or is that just what is used in this study? I would consider modifying this sentence to say something like "Typically, . . ." or "In this study, . . .".

3. P3 L13-L14: This sentence needs some clarification, as numerous studies have already been described that employ triple-frequency radar retrievals. Can the exact novel aspect of this study's triple-frequency radar retrieval be stated more clearly here?

4. P4 L16-17: This sentence is a bit confusing. Given the description of the assumptions on how each particle is treated (e.g., as a homogenous spheroid) when calculating the radar backscatter cross-sections, it might be clearer to state something along the lines of, "Approximations of the microphysical structure are used to calculate the radar backscatter cross-section", or, "The microphysical structure is represented through an approximation when calculating the radar backscatter cross-section".

5. Figure 2: There should be boxes around the legends, particularly in panel (b) to differentiate it from actual data points. It should also be clarified that the y-axis units in (a) are in linear units (or convert them to log units) since it was previously stated DWR

would be expressed in dB. Finally, the y-axis labels have no context – what is f? For (a), either explain in the caption or just convert it to dB, since DWR has already been explained in the text, and for (b) perhaps just state "Volume-Weighted Concentration".

6. P9 L29: ". . .. The many narrow features of the backscatter cross-section ratio spectra are smoothed out" should have "when integrated across the PSD" added to it.

7. P14 L1-L4: I was initially confused with how this differed from the analysis presented in Figure 1, but gathered later that the PSD shape factor and density factor used were informed by the observed precipitation. I suggest making that fact more explicit and putting it earlier in the section.

8. P14 L10: It looks to me like the frontal snow regime has DWR10-35 exceeding 10 dB?

9. P16 L24: The sentence "the state vector is linearly interpolating between the re-trieved state vectors at the retrieved value of PSD shape. . ." is unclear to me. Once the optimal $\mu$ value is found, which state vectors is the "retrieved" state vector interpolated between? Also, should "found by" be before "linearly interpolating"?

Technical Corrections:

1. P1 L1: "cloud" should be "clouds"

2. P2 L26; P12 L8; P12 L20: "remote-sensed" should be "remotely-sensed"

3. P2 L26-L28: This sentence should be modified to be, ". . . in-situ measurements of snow events (Kneifel et al., 2015), but more detail of how the parameters. . .. remains to be explored."

4. P2 L31-L32: No hyphens are needed between the frequency and unit as they are not acting as compound descriptors in this context. 95 should also have a 'GHz' after it, and I'd add a comma after "35 GHz".

5. P3 L10-13: These sentences should be reworked avoid three separate clauses

strung together with semicolons. Consider making the third clause its own sentence.

6. P4 L16: "in" should be "as".

7. P6 L6: "95-GHz" should be "95 GHz".

8. P6 L20: "assumptions to" should be "assumptions of".

9. P7 L8: "(Kneifel et al., 2015)" should be "Kneifel et al. (2015)".

10. P7 L25: I believe this semicolon should be a comma.

11. P9 L14-L15: I would be consistent and just refer to "the ratios between radar backscatter cross-sections at 10-35-GHz and 35-95-GHz" as the DWR10-35 and DWR35-95 as already done in the text. This applies to the legend and caption of Figure 2 as well.

12. P16 L27: "The retrieved timeseries of PSD..." should be "The retrieved timeseries of PSD shape..."

13. P16 L29: "...the retrieved PSD noisy..." should be "... the retrieved PSD shape [factor] is noisy..."

14. P17 L24: "mixed-phase cloud" should be "mixed-phase clouds".

15. P17 L25: Should "distribution" be "relation"?

16. P18 L5: A space is needed after "surface".

17. P19 L1: "case comprised compact graupel..." should be "case was comprised of compact graupel..."

---

## Referee Comment (RC2) · Anonymous Referee #2 · 28 Apr 2019

Review of amt-2019-100

General comments:

This study investigates the effects of PSD parameters and ice particle morphology on the triple-frequency radar signatures of snow. It introduces a novel methodology for the triple frequency Doppler radar retrieval of snow PSD parameters. Overall, the content of the manuscript is within the scope of the AMT journal. However, there is a concern about the short duration of the rimed vs. aggregated snow periods, where only one case is utilized in the radar analysis. This may not be sufficient for generalizing the results because of the absence of more robust rimed/aggregated radar-based statistics/measurements. If this paper is meant to be a single case study, it should be stated clearly from the beginning, and especially in the conclusion. Reading the paper, one can easily get an impression that the main findings of the study are potentially universal, but the generalization of the results should not be based off of a one case. Major revision is recommended.

Specific comments:

The greatest concern is about the representativeness of the measurements obtained from the 10 minutes of rimed and 15 minutes of aggregated snow from the case study. This may not be sufficient to draw generalized conclusions about how this approach and overall novel methodology works. Slightly different environmental conditions could potentially produce altered results. The recommendation is to increase the number of cases for your radar analysis, perhaps 4-5 should suffice. Measurements from different geographical/climatological regions could also help to solidify your findings. If there is not much difference between the updated and the findings from the current version of the manuscript, add few paragraphs and/or table describing the statistics of the new dataset and retain the rest of the current analysis. If large discrepancies occur, the suggestion is to present a case with the statistics close to the one obtained from all available measurements. In this way, the generalization of the results would be justified.

Technical corrections:

P7 L8: (Kneifel et al., 2015) should be Kneifel et al. (2015)

Figure 4: Add the temperature contours to the image if available.

Figure 9: "PSD shape $\mu$" should be "PSD shape parameter $\mu$"

P21 L10-14: This sentence is a bit hard to follow, perhaps split it in two.

---

## Referee Comment (RC3) · Anonymous Referee #3 · 1 May 2019

Review of AMTD – amt-2019-100

The importance of particle size distribution shape for triple-frequency radar retrievals of the morphology of snow

Overall this is a well-written paper and an interesting study that falls within the scope of the AMT journal. The work focuses on assessing the quality of triple-frequency retrievals with specific aims at better constraints on snow particle characteristics – especially with regard to snow particle size distribution (PSD). This work builds from

previous triple-frequency work with nice results contrasting rimed and aggregate snow particles. In general, I have only minor comments and some technical suggestions.

General Comments:

* Why do you use the mean Doppler velocity at 35 GHz in the retrieval? I was unable to find why this frequency is optimal for the retrieval methodology. Please add some information.

* PIP PSDs and other products are available at 1 min resolution. Why do you choose 5 min? If it is for better statistics through averaging, please make that point. Also, if you are using these values during the 25 minutes of the event, this essentially leaves 5 points for comparison with the retrieval. Could you expand on why you feel this is enough in situ data for assessment of aspects of the retrieval?

* How did the prefrontal versus frontal period get defined? Is it purely from radar features? Maybe something more rigorous or not from the radar would be more appropriate (since you are using the radar to evaluate the method). Is there collocated met equipment that can be used to determine the onset of the front? You have collocated radiosondes – are those used to determine the timing of the front? Or maybe could help justify the time chosen.

* Since one event is being used to test the efficacy of this method, I think this needs to be emphasized. Also, would be good to argue why this one event may be applicable to other similar particles or events in different locations.

Specific Comments:

Page 2, Lines 7 – 11: Split into two sentences

Page 2, Lines 26 – 27: "but it remains to explore…" does not make sense

Page 2, Line 31: Be consistent with your "-" or not for frequencies (either all should be 35-GHz or should be 35 GHz

Page 3, Line 12: replace the "measurements;" with "measurements."

Page 3, Line 14: "advantages" should be "advantage"

Page 4, Line 1: Define CAPTIVATE

Page 4, Line 23: Make mass-size equation on own line with equation number

Page 4, Line 27: "AR" should be in parentheses. Also, the author goes between saying AR, axial ratio, and aspect ratio throughout the document. Be consistent (I recommend "AR" since you define it)

Page 5, Line 5: Should measurement vectors be numbered as well?

Page 7, Line 28: aspect ratio "AR"

Page 7, Line 34: "The range of radar signatures is overlaid with the measured triple-frequency radar data from Hyytiälä…" You already talk about the shape of the data (the hook feature) earlier. I think you should introduce the overlaying of this data before getting into the description above.

Page 9, Line 2: 3mm for the spheroids but maybe more like 4mm for the fractal particles?

Page 9, Lines 10 – 11: Reference needed.

Page 9, Line 34: "We may therefore…"

Page 12, Lines 3 – 5: Please add some details about the frontal passage – i.e., met data or observations that are not radar or particle focused. This will justify better your distinction of the two regimes (since you are using radar and in situ particle obs to test and assess the retrieval).

Page 12, Lines 20 – 22: Reference needed

Page 14, Lines 2 – 3: "…suggesting that some rimed particles persist after 23:03
[Figure]

UTC."

Could it be the choice of timing of the prefrontal versus frontal is off? Would using collocated met data clarify this?

Page 14, Line 7: ". . .in situ measurements at the surface." Please specify what this is from – I assume the PIP measurements?

Page 18, Lines 12 – 13: I do not understand "A significant difference between the frontal and the prefrontal profiles is that all retrievals are able to represent the observed profile of mean Doppler velocity below about 1.5 km," as I am not see this. Could you please add some details as to what you are referring?

Page 19, Lines 9 – 11: The truncation you refer to here – are you talking about the PIP or the method? Both have lower limits. And technically snow is always dominated by small particles – just less so or more so depending on the shape of the PSD. So I do not think this is the correct sentiment here (i.e., even when a PSD is quite broad with lots of large aggregates, there still tends to be 1 to 2 orders of magnitude more small particles. When the PSD is quite narrow that is more like a factor of 3 to 4 orders of mag. . . but still lots of small particles in a broad distribution).

Pages 19 – 23: This section almost feels a bit out of order. It is like there are conclusions at the beginning and the discussion of application shown in Fig. 9 further into the section. It may help potential readers to move the discussion of the Fig. 9 to earlier in this section and move the verbiage in the beginning of the section to later – as a transition to conclusions.

---

## Author Comment (AC1) · 5 Aug 2019

We thank the reviewers for their constructive feedback on the manuscript. A recurring suggestion was that we apply the retrieval to more data than the 25-minute case study from 21 February 2014 originally used. We agree that this is desirable, reiterating that while the colocated remotely-sensed and in situ measurements of snow from BAECC 2014 are extremely valuable and of a high quality, the number of cases are limited. A related comment was that we should more clearly acknowledge the limited measurement period to which our retrieval was applied. In addressing both of these suggestions, we have both included an additional case study and de-emphasised the

retrieval of the PSD shape parameter in discussing the results.

During the snow experiment intensive observation period of BAECC there were three cases in which all three radars were zenith-pointing during a snow event, and where the snowfall at the surface was not affected by melting (the cases shown in Kneifel et al. 2015). The snowfall at the surface during one of these cases (7 February 2014) was insufficient for the in situ snow retrieval of von Lerber et al. (2015). We have therefore expanded our study to include 60 minutes of snowfall from the 16 February 2014 case. This case also includes riming, but is notable for the presence of secondary ice production due to rime splintering (the Hallett-Mossop process). These secondary needles rapidly aggregate, such that the radar measurements in this case are dominated by large aggregate snowflakes with a very open structure, while the in situ measurements include a mixture of graupel, large aggregates, and needles.

In applying our retrieval to this case, it was evident from PIP measurements that the PSD shape was nearly constant, but that significant changes in the triple-frequency radar signature could be attributed to the presence of large aggregates of needles, consistent with the findings of Leinonen and Moisseev (2015). We have therefore expanded the scope of the study to include the effects of variations in the internal structure of aggregates, which are represented within the SSRGA. We hope the reviewers agree that expanding the study to address both the PSD shape parameter and the internal structure of aggregates strengthens this effort to better understand and interpret the parameters affecting triple-frequency radar measurements.

To summarise, we have made the following changes:

- The title is now, "The importance of particle size distribution and internal structure for triple-frequency radar retrievals of the morphology of snow"

- We added a coauthor, Leonie von Terzi at the University of Cologne.

- We expanded our discussion of the coefficients of the SSRGA, especially those

relating to the internal structure of aggregate particles, in Sections 2.1 and 3. L. von Terzi's contribution to the study was to perform simulations of aggregation of various monomers and their SSRGA coefficients; we use this to identify aggregates of needles as having triple-frequency radar signatures with especially low values of $\mathrm{DWR}_{35-95}$ compared to aggregates of other monomers.

- Section 5 now uses the 16 February case study to explore triple-frequency radar measurements and retrievals from a case featuring rime splintering. This is a very distinct situation from the first case study, and combined the two cases cover the wide range of triple-frequency radar measurements from during BAECC 2014.

- The discussion and conclusions (Section 6) have been substantially re-written to be more concise, while addressing the expanded scope of the paper.

**Reviewer #1**

Specific comments

**This may be personal preference, but throughout the paper "PSD shape" is used to refer to the parameter $\mu$–while I understand in the normalized distribution space $\mu$ does modify the actual shape (width) of the distribution, when talking about $\mu$ I think it may be clearer to refer to this as the "PSD shape factor" or "PSD shape parameter", as done on P4 L9.**

We agree that this is most consistent and clear, and now refer everywhere to "PSD shape parameter".

**P2 L30-L32: Are triple-frequency measurements always at precisely 95 GHz,**

**35GHz, and a third frequency below 15 GHz? Or is that just what is used in this study? I would consider modifying this sentence to say something like "Typically,..." or "In this study,...".**

Good point. In the literature on triple-frequency signatures of snow this configuration of radars is typical. However, it's true that the lowest frequency radar have been any of C, X or Ku-bands radars. The sentence now reads:

> The triple-frequency radar 'signature' consists of radar measurements at three frequencies spanning the millimeter wavelength range, the two dual-wavelength ratios (DWRs) derived from radar measurements of which reveal information about non-Rayleigh scattering from larger snowflakes. Typically radars at 95, 35-GHz 35 and a third frequency below 15-GHz, and provides between 3 and 15 GHz are used [e.g. *Kneifel et al.*, 2015; *Leinonen and Szyrmer*, 2015; *Barrett et al.*, 2019]

**P3 L13-L14: This sentence needs some clarification, as numerous studies have already been described that employ triple-frequency radar retrievals. Can the exact novel aspect of this study's triple-frequency radar retrieval be stated more clearly here?**

Indeed, the novelty is that both the triple-frequency radar reflectivity factors and mean Doppler velocity measurements are assimilated in order to estimate different aspects of the particle morphology. This sentence now reads,

> To our knowledge , a a retrieval assimilating both triple-frequency Doppler retrieval of ice radar reflectivity factors and mean Doppler velocity to estimate the properties of snow has not yet been described—this may described. This approach should have the advantages of estimating particle

density constrained by constraining particle density with Doppler velocity [as in *Mason et al.*, 2018]and , while using triple-frequency radar signatures to constrain some parameters of particle morphologyadditional parameter affecting the microphysical properties or size distribution of precipitating ice particles.

**P4 L16-17: This sentence is a bit confusing. Given the description of the assumptions on how each particle is treated (e.g., as a homogenous spheroid) when calculating the radar backscatter cross-sections, it might be clearer to state something along the lines of, "Approximations of the microphysical structure are used to calculate the radar backscatter cross-section", or, "The microphysical structure is represented through an approximation when calculating the radar backscatter cross-section".**

We agree. The sentence now reads,

> The morphology of the ice particles is represented by three parameters controlling their microphysical structure, density, and shape. The microphysical structure is represented in an approximation to Approximations to the microphysical structure of ice particles are used to calculate the radar backscatter cross-section $\sigma(D)$ ($\sigma$) of each particle.

**Figure 2: There should be boxes around the legends, particularly in panel (b) to differentiate it from actual data points. It should also be clarified that the y-axis units in(a) are in linear units (or convert them to log units) since it was previously stated DWR would be expressed in dB. Finally, the y-axis labels have no context – what is f? For(a), either explain in the caption or just convert it to**

**dB, since DWR has already been explained in the text, and for (b) perhaps just state "Volume-Weighted Concentration".**

We have modified the titles and y-axis labels of both subplots to make the quantities clearer, and have added an equation to define the dual-backscatter ratio (DBR) consistent with Kneifel et al. 2016, and which is now shown in dB. Boxes have been also added to the legends in plot (b).

**P9 L29: "... The many narrow features of the backscatter cross-section ratio spectra are smoothed out" should have "when integrated across the PSD" added to it.**

This is indeed clearer. We've made the change.

**P14 L1-L4: I was initially confused with how this differed from the analysis presented in Figure 1, but gathered later that the PSD shape factor and density factor used were informed by the observed precipitation. I suggest making that fact more explicit and putting it earlier in the section.**

Thank you, this was helpful feedback. To better highlight the important information in each figure, we no longer overlay the triple-frequency signatures with triple-frequency radar measurements in the first figure. This makes it easier in this figure to interpret how the triple-frequency radar signatures for different particle types change with the physical parameters, and leaves the comparison with measurements to the case studies (Figs. 7 & 12).

**P14 L10: It looks to me like the frontal snow regime has DWR10-35 exceeding 10dB?**

Thank you, we've made this change.

**P16 L24: The sentence "the state vector is linearly interpolating between the retrieved state vectors at the retrieved value of PSD shape..." is unclear to me. Once the optimal $\mu$ value is found, which state vectors is the "retrieved" state vector interpolated between? Also, should "found by" be before "linearly interpolating"?**

This sentence was not only unclear, but incorrect: we simply take the retrieved state vector from the retrieval that minimises the errors in DWR. This is now explained in the text.

> We carry out a pseudo-retrieval by running multiple $Z_{10,35,95}\,V_{35}$ retrievals in which PSD shape is assumed to take the PSD shape parameter takes integer values from $\mu = -2$ to $\mu = 10$. The pseudo-retrieval is made by selecting the value of PSD shape that retrieved value is that which minimises the error in forward-modelled DWR$_{35-95}$ and DWR$_{10-35}$ between 400 and 600 m above ground level (Fig. 10a & b); the state vector is linearly interpolating between the retrieved state vectors at the retrieved value . The forward-modelled DWRs for a range of PSD shape , giving an estimate of the other retrieved quantities when the PSD shape is retrieved (Fig. 10parameters span up to 4 d–g). dB, with the range depending on the median volume diameter. To reduce noise in the pseudo-retrieval the minimisation is carried out at a smoothed temporal resolution of 15 s.

Technical Corrections:

Thank you for your thorough reading. We have gratefully made all of the suggested changes.

[Figure]

**P1 L1:** "cloud" should be "clouds"

**P2 L26; P12 L8; P12 L20:** "remote-sensed" should be "remotely-sensed"

**P2 L26-L28:** This sentence should be modified to be, "... in-situ measurements of snow events (Kneifel et al., 2015), but more detail of how the parameters... remains to be explored."

**P2 L31-L32:** No hyphens are needed between the frequency and unit as they are not acting as compound descriptors in this context. 95 should also have a 'GHz' after it, and I'd add a comma after "35 GHz".

**P3 L10-13:** These sentences should be reworked avoid three separate clauses strung together with semicolons. Consider making the third clause its own sentence.

**P4 L16:** "in" should be "as".

**P6 L6:** "95-GHz" should be "95 GHz".

**P6 L20:** "assumptions to" should be "assumptions of".

**P7 L8:** "(Kneifel et al., 2015)" should be "Kneifel et al. (2015)".

**P7 L25:** I believe this semicolon should be a comma.

**P9 L14-L15:** I would be consistent and just refer to "the ratios between radar backscatter cross-sections at 10-35-GHz and 35-95-GHz" as the DWR10-35 and DWR35-95 as already done in the text. This applies to the legend and caption of Figure 2 as well.

**P16 L27:** "The retrieved timeseries of PSD..." should be "The retrieved time-series of PSD shape..."

**P16 L29:** "...the retrieved PSD noisy..." should be "... the retrieved PSD shape [factor] is noisy..."

**P17 L24:** "mixed-phase cloud" should be "mixed-phase clouds".

**P17 L25:** Should "distribution" be "relation"?

**P18 L5:** A space is needed after "surface".

**P19 L1:** "case comprised compact graupel..." should be "case was comprised of compact graupel..."

---

## Author Comment (AC2) · 5 Aug 2019

We thank the reviewers for their constructive feedback on the manuscript. A recurring suggestion was that we apply the retrieval to more data than the 25-minute case study from 21 February 2014 originally used. We agree that this is desirable, reiterating that while the colocated remotely-sensed and in situ measurements of snow from BAECC 2014 are extremely valuable and of a high quality, the number of cases are limited. A related comment was that we should more clearly acknowledge the limited measurement period to which our retrieval was applied. In addressing both of these suggestions, we have both included an additional case study and de-emphasised the

retrieval of the PSD shape parameter in discussing the results.

During the snow experiment intensive observation period of BAECC there were three cases in which all three radars were zenith-pointing during a snow event, and where the snowfall at the surface was not affected by melting (the cases shown in Kneifel et al. 2015). The snowfall at the surface during one of these cases (7 February 2014) was insufficient for the in situ snow retrieval of von Lerber et al. (2015). We have therefore expanded our study to include 60 minutes of snowfall from the 16 February 2014 case. This case also includes riming, but is notable for the presence of secondary ice production due to rime splintering (the Hallett-Mossop process). These secondary needles rapidly aggregate, such that the radar measurements in this case are dominated by large aggregate snowflakes with a very open structure, while the in situ measurements include a mixture of graupel, large aggregates, and needles.

In applying our retrieval to this case, it was evident from PIP measurements that the PSD shape was nearly constant, but that significant changes in the triple-frequency radar signature could be attributed to the presence of large aggregates of needles, consistent with the findings of Leinonen and Moisseev (2015). We have therefore expanded the scope of the study to include the effects of variations in the internal structure of aggregates, which are represented within the SSRGA. We hope the reviewers agree that expanding the study to address both the PSD shape parameter and the internal structure of aggregates strengthens this effort to better understand and interpret the parameters affecting triple-frequency radar measurements.

To summarise, we have made the following changes:

- The title is now, "The importance of particle size distribution and internal structure for triple-frequency radar retrievals of the morphology of snow"

- We added a coauthor, Leonie von Terzi at the University of Cologne.

- We expanded our discussion of the coefficients of the SSRGA, especially those

relating to the internal structure of aggregate particles, in Sections 2.1 and 3. L. von Terzi's contribution to the study was to perform simulations of aggregation of various monomers and their SSRGA coefficients; we use this to identify aggregates of needles as having triple-frequency radar signatures with especially low values of $\mathrm{DWR}_{35-95}$ compared to aggregates of other monomers.

- Section 5 now uses the 16 February case study to explore triple-frequency radar measurements and retrievals from a case featuring rime splintering. This is a very distinct situation from the first case study, and combined the two cases cover the wide range of triple-frequency radar measurements from during BAECC 2014.

- The discussion and conclusions (Section 6) have been substantially re-written to be more concise, while addressing the expanded scope of the paper.

**Specific comments:**

**The greatest concern is about the representativeness of the measurements obtained from the 10 minutes of rimed and 15 minutes of aggregated snow from the case study. This may not be sufficient to draw generalized conclusions about how this approach and overall novel methodology works. Slightly different environmental conditions could potentially produce altered results. The recommendation is to increase the number of cases for your radar analysis, perhaps 4-5 should suffice. Measurements from different geographical/climatological regions could also help to solidify your findings. If there is not much difference between the updated and the findings from the current version of the manuscript, add few paragraphs and/or table describing the statistics of the new dataset and retain the rest of the current analysis. If large discrepancies occur, the suggestion is to present a case with the statistics close to the one obtained from all**

**available measurements. In this way, the generalization of the results would be justified.**

As addressed in our general comments above, unfortunately a further 4 or 5 suitable snow events were not measured during BAECC 2014, but we have expanded the study to include a second, longer case study in which the snowfall differs significantly from the 21 February case. We take the broader point that our results for these case studies are not necessarily generalizable: the two contrasting case studies help to demonstrate this, and we have substantially re-written our discussion and conclusions to better represent the remaining uncertainties.

**Technical corrections**

We have gratefully made the following changes:

**P7 L8: (Kneifel et al., 2015) should be Kneifel et al. (2015)**

**Figure 4: Add the temperature contours to the image if available.**

**Figure 9: "PSD shape $\mu$" should be "PSD shape parameter $\mu$"**

**P21 L10-14: This sentence is a bit hard to follow, perhaps split it in two.**

---

## Author Comment (AC3) · 5 Aug 2019

We thank the reviewers for their constructive feedback on the manuscript. A recurring suggestion was that we apply the retrieval to more data than the 25-minute case study from 21 February 2014 originally used. We agree that this is desirable, reiterating that while the colocated remotely-sensed and in situ measurements of snow from BAECC 2014 are extremely valuable and of a high quality, the number of cases are limited. A related comment was that we should more clearly acknowledge the limited measurement period to which our retrieval was applied. In addressing both of these suggestions, we have both included an additional case study and de-emphasised the

retrieval of the PSD shape parameter in discussing the results.

During the snow experiment intensive observation period of BAECC there were three cases in which all three radars were zenith-pointing during a snow event, and where the snowfall at the surface was not affected by melting (the cases shown in Kneifel et al. 2015). The snowfall at the surface during one of these cases (7 February 2014) was insufficient for the in situ snow retrieval of von Lerber et al. (2015). We have therefore expanded our study to include 60 minutes of snowfall from the 16 February 2014 case. This case also includes riming, but is notable for the presence of secondary ice production due to rime splintering (the Hallett-Mossop process). These secondary needles rapidly aggregate, such that the radar measurements in this case are dominated by large aggregate snowflakes with a very open structure, while the in situ measurements include a mixture of graupel, large aggregates, and needles.

In applying our retrieval to this case, it was evident from PIP measurements that the PSD shape was nearly constant, but that significant changes in the triple-frequency radar signature could be attributed to the presence of large aggregates of needles, consistent with the findings of Leinonen and Moisseev (2015). We have therefore expanded the scope of the study to include the effects of variations in the internal structure of aggregates, which are represented within the SSRGA. We hope the reviewers agree that expanding the study to address both the PSD shape parameter and the internal structure of aggregates strengthens this effort to better understand and interpret the parameters affecting triple-frequency radar measurements.

To summarise, we have made the following changes:

- The title is now, "The importance of particle size distribution and internal structure for triple-frequency radar retrievals of the morphology of snow"

- We added a coauthor, Leonie von Terzi at the University of Cologne.

- We expanded our discussion of the coefficients of the SSRGA, especially those

relating to the internal structure of aggregate particles, in Sections 2.1 and 3. L. von Terzi's contribution to the study was to perform simulations of aggregation of various monomers and their SSRGA coefficients; we use this to identify aggregates of needles as having triple-frequency radar signatures with especially low values of $DWR_{35-95}$ compared to aggregates of other monomers.

- Section 5 now uses the 16 February case study to explore triple-frequency radar measurements and retrievals from a case featuring rime splintering. This is a very distinct situation from the first case study, and combined the two cases cover the wide range of triple-frequency radar measurements from during BAECC 2014.

- The discussion and conclusions (Section 6) have been substantially re-written to be more concise, while addressing the expanded scope of the paper.

**General comments**

**Why do you use the mean Doppler velocity at 35 GHz in the retrieval? I was unable to find why this frequency is optimal for the retrieval methodology. Please add some information.**

We now note in both Section 2.2 and in the case study and retrieval (Section 4) that the 95 GHz radar had a mispointing error during BAECC 2014 that makes it difficult to use the mean Doppler velocity. We therefore use the Doppler velocity from the most sensitive available instrument, the 35 GHz. In practice for snow near the surface, the 10 GHz mean Doppler velocity could also have been used.

**PIP PSDs and other products are available at 1 min resolution. Why do you choose 5 min? If it is for better statistics through averaging, please make that**

[Figure]

**point. Also, if you are using these values during the 25 minutes of the event, this essentially leaves 5 points for comparison with the retrieval. Could you expand on why you feel this is enough in situ data for assessment of aspects of the retrieval?**

We are not using the PIP data directly, but rather the retrieval of von Lerber et al. (2016), which includes an estimate of the snow bulk density. This method requires a sufficient sample of ice particles, hence the use of a 5 minute resolution. We have now added this explanation to Section 2.2:

> In situ measurements of snow at the surface are provided by the Particle Imaging Package (PIP) video disdrometer (Newman et al. 2009). While the temporal resolution of PIP measurements is 1 minute, estimates of parameters of the PSD and particle properties are made over 5 minute intervals in order to increase the statistical sampling during BAEC while still resolving changes in the properties of snowfall at the surface, as described in von Lerber et al. (2014) (also Moisseev et al. 2017, Tiira et al. 2016. The method of moments is used to estimate the parameters of the Gamma distribution from the measured PSD (Moisseev and Chandrasekar, 2007).

**How did the prefrontal versus frontal period get defined? Is it purely from radar features? Maybe something more rigorous or not from the radar would be more appropriate (since you are using the radar to evaluate the method). Is there collocated met equipment that can be used to determine the onset of the front? You have collocated radiosondes – are those used to determine the timing of the front? Or maybe could help justify the time chosen.**

We agree that the definition of the front was not clear, and in fact was probably not necessary to make the analysis, which is more focused on comparing the rimed and

unrimed snow. We now refer to the two regimes in this case as "rimed" and "unrimed", rather than "prefrontal" and "frontal", based on the transition in radar and particle imaging measurements (Figs. 6 & 7).

**Since one event is being used to test the efficacy of this method, I think this needs to be emphasized. Also, would be good to argue why this one event may be applicable to other similar particles or events in different locations.**

We agree. We now more strongly stress that the limited cases from BAECC 2014 help to demonstrate the effects of the PSD shape parameter and internal structure of aggregates on triple-frequency radar measurements, but that longer and more diverse measurements are needed to better understand their relative importance to global snowfall.

**Specific comments**

Unless responded to directly, we have appreciatively made the following changes.

**Page 2, Lines 7 – 11: Split into two sentences**

**Page 2, Lines 26 – 27: "but it remains to explore..." does not make sense**

**Page 2, Line 31: Be consistent with your "-" or not for frequencies (either all should be 35-GHz or should be 35 GHz**

**Page 3, Line 12: replace the "measurements;" with "measurements."**

**Page 3, Line 14: "advantages" should be "advantage"**

**Page 4, Line 1: Define CAPTIVATE**

**Page 4, Line 23:** **Make mass-size equation on own line with equation number**

**Page 4, Line 27:** **"AR" should be in parentheses. Also, the author goes between saying AR, axial ratio, and aspect ratio throughout the document. Be consistent (I recommend "AR" since you define it)**

**Page 5, Line 5:** **Should measurement vectors be numbered as well?**

**Page 7, Line 28:** **aspect ratio "AR"**

**Page 7, Line 34:** **"The range of radar signatures is overlaid with the measured triple-frequency radar data from Hyytiälä..." You already talk about the shape of the data (the hook feature) earlier. I think you should introduce the overlaying of this data before getting into the description above.**

In response to this and other comments, we have removed the overlaid radar measurements from these figures (as well as adding a second kind of aggregate snowflake model). This makes the main focus of this figure the sensitivity to the different parameters, and also shows more clearly the fit to observations when we show the case studies. The discussion has been modified accordingly, and is hopefully now more linear.

**Page 9, Line 2:** **3mm for the spheroids but maybe more like 4mm for the fractal particles?**

We now just say "around 3mm".

**Page 9, Lines 10 - 11:** **Reference needed.**

This was the result shown in the figure. To better link the earlier result, we now write,

> We have shown that, based on simulated radar backscatter cross-sections, the PSD shape parameter has a greater influence on the

simulated triple-frequency radar signature than the well-known effect
of particle density.

**Page 9, Line 34: "We may therefore..."**

**Page 12, Lines 3 - 5: Please add some details about the frontal passage – i.e., met
data or observations that are not radar or particle focused. This will justify
better your distinction of the two regimes (since you are using radar and in
situ particle obs to test and assess the retrieval).**

As stated above: we have chosen to de-emphasise the frontal passage in this
case, as it is not strictly necessary for interrogating the snow microphysics. We
have added a reference to the more detailed dicussion in Kneifel et al. 2015.

**Page 12, Lines 20 - 22: Reference needed**

**Page 14, Lines 2 - 3: "...suggesting that some rimed particles persist after 23:03
UTC." Could it be the choice of timing of the prefrontal versus frontal is
off? Would using collocated met data clarify this?**

As discussed earlier, we now simply distinguish between rimed and unrimed
snow regimes. In this context, it doesn't seem especially problematic that there
would be a mixture of rimed and unrimed snow in the transition between the
regimes.

**Page 14, Line 7: "...in situ measurements at the surface." Please specify what this
is from—I assume the PIP measurements?**

**Page 18, Lines 12 —13: I do not understand "A significant difference between the
frontal and the prefrontal profiles is that all retrievals are able to represent
the observed profile of mean Doppler velocity below about 1.5 km," as I
am not see this. Could you please add some details as to what you are
referring?**

We agree that this was unclear. This sentence now reads,

> A significant difference between the profiles is that in the unrimed
> regime retrievals that do not assimilate Doppler velocity are able to rep-
> resent the observed profile of mean Doppler velocity: this is because
> the a priori density factor ($r = 0$) makes a reasonable estimate of the
> terminal fallspeed of unrimed aggregate particles, provided that their
> size is well-constrained by dual-frequency radar measurements.

**Page 19, Lines 9 – 11: The truncation you refer to here – are you talking about the PIP or the method? Both have lower limits. And technically snow is always dominated by small particles – just less so or more so depending on the shape of the PSD. So I do not think this is the correct sentiment here (i.e., even when a PSD is quite broad with lots of large aggregates, there still tends to be 1 to 2 orders of magnitude more small particles. When the PSD is quite narrow that is more like a factor of 3 to 4 orders of mag... but still lots of small particles in a broad distribution).**

We now cite Moisseev and Chandresekhar (2007) regarding the effects of dis-
drometer truncation on the method of moments.

**Pages 19 – 23: This section almost feels a bit out of order. It is like there are con-
clusions at the beginning and the discussion of application shown in Fig.
9 further into the section. It may help potential readers to move the dis-
cussion of the Fig. 9 to earlier in this section and move the verbiage in the
beginning of the section to later – as a transition to conclusions.**

Thank you for this—we agree, and hope that the discussion and conclusions have
benefited from a significant re-write. We have opted to remove Fig.9 after intro-
ducing additional figures for the second case study, while still briefly discussing
that it is unknown how closely related the PSD shape parameter is with riming.

Hopefully the flow of the discussion and conclusions is now clearer and more concise.

---

## Referee Report (RR1)

Review of amt-2019-100-version3

The manuscript is improved from the previous version. Minor, mostly technical revision is recommended.

Page 11, Figure 3: The notation is slightly off, fractal particles are in panels (a-f) and homogeneous spheroids are in (g-i).

Page 16, line 5: The "$DWR_{35-94}$" should be "$DWR_{35-95}$".

Page 19, line 12: The "affect" should be "affected".

Page 19, line 16: Rearrange the order of words, "measurements assimilated" should be "assimilated measurements".

Page 25, lines 10 and 31: You probably meant to reference Fig. 11 (11d) instead of Fig. 5 (5d).

Page 27, lines 5-6: Fig. 13c is the PSD shape parameter, you meant Fig. 13e, which is $D_0$. Also I don't see which lower values of $DWR_{35,95}$ you are referring to, with respect to what? Add some range to the statement. Be consistent with the previous notation, "$DWR_{35,95}$" should be "$DWR_{35-95}$".

Page 28, line 12: Fig. 13 should be Fig 13b.

Page 29, Figure 13: 10-94 GHZ should be 10-95 GHz, and also 10-35 GHx should be 10-35 GHz. Also, the PSD shape parameter μ should be -1 instead of 1.